# Diamond with $Sp^2$-$Sp^3$ composite phase for thermometry at Millikelvin temperatures

Jianan Yin[1,2,3,9], Yang Yan[1,2,3,9], Mulin Miao[3,4], Jiayin Tang[5], Jiali Jiang[2,3], Hui Liu[1,2,3], Yuhan Chen[2,3], Yinxian Chen[3,4], Fucong Lyu [1,2,3], Zhengyi Mao [1,2,3], Yunhu He[1,2,3], Lei Wan[1,2,3,6], Binbin Zhou[7] & Jian Lu [1,2,3,4,8] ✉

Temperature is one of the seven fundamental physical quantities. The ability to measure temperatures approaching absolute zero has driven numerous advances in low-temperature physics and quantum physics. Currently, milli-kelvin temperatures and below are measured through the characterization of a certain thermal state of the system as there is no traditional thermometer capable of measuring temperatures at such low levels. In this study, we develop a kind of diamond with $sp^2$-$sp^3$ composite phase to tackle this problem. The synthesized composite phase diamond (CPD) exhibits a negative temperature coefficient, providing an excellent fit across a broad temperature range, and reaching a temperature measurement limit of 1 mK. Additionally, the CPD demonstrates low magnetic field sensitivity and excellent thermal stability, and can be fabricated into probes down to 1 micron in diameter, making it a promising candidate for the manufacture of next-generation cryogenic temperature sensors. This development is significant for the low-temperature physics researches, and can help facilitate the transition of quantum computing, quantum simulation, and other related technologies from research to practical applications.

Low-temperature physics is a critical field that can help improve our understanding of the universe, matter, and technology[1–3]. At temperatures close to absolute zero, matter exhibits a range of unusual and exotic properties that cannot be observed at higher temperatures; these properties are important in several related fields such as quantum computing, quantum simulation, superfluidity, and Bose-Einstein condensation[1,4–11]. Although quantum gases with a temperature of 38 pK have already been produced in the laboratory[12], traditional thermometers can only measure temperatures higher than 20 mK[13]. Lowering the temperature measurement limit is important, particularly for practical applications of various technologies in the fields of low-temperature physics and quantum physics.

A few $sp^2$-hybridized carbon materials have shown good potential for cryogenic temperature measurements[14]. Compared to diamond, graphite has a softer texture and inferior mechanical properties, making it highly susceptible to flaking or getting damaged during use. The extraordinarily low coefficient of thermal expansion, exceptional hardness, wide band gap, and high hole mobility of diamond render it an ideal material for temperature measurement in extreme environments[15–18]. Inspired by our previous work[19,20], we designed the diamond with $sp^2$-$sp^3$ composite phase to overcome the bottleneck of

[1]CityU-Shenzhen Futian Research Institute, Shenzhen 518045, China. [2]Department of Mechanical Engineering, City University of Hong Kong, Hong Kong, China. [3]Hong Kong Branch of National Precious Metals Material Engineering Research Centre, City University of Hong Kong, Hong Kong, China. [4]Department of Materials Science and Engineering, City University of Hong Kong, Hong Kong, China. [5]Department of Physics, City University of Hong Kong, Hong Kong, China. [6]China Resources Research Institute of Science and Technology Co, Limited, Hong Kong, China. [7]Shenzhen Institute of Advanced Electronic Materials, Shenzhen Institute of Advanced Technology, Chinese Academy of Sciences, Shenzhen, China. [8]Centre for Advanced Structural Materials, City University of Hong Kong Shenzhen Research Institute, Greater Bay Joint Division, Shenyang National Laboratory for Materials Science, Shenzhen, China. [9]These authors contributed equally: Jianan Yin, Yang Yan. ✉e-mail: jianlu@cityu.edu.hk

low-temperature measurement. Dual-phase structures have been widely employed to enhance the mechanical properties of metals and ceramics[19,21,22]. Several researchers have investigated carbon-based materials with dual-phase or multi-level nanostructures and improved their mechanical or electrical properties[20,23–27]. Furthermore, the combination of $sp^2$-hybridized phases with diamond can significantly impacts the electrical conductivity, bandgap, and other characteristics of diamond-based materials[28,29]. This approach offers insights to the potential applications of diamond in the field of electronics. In this work, in order to differentiate from the cubic-hexagonal dual-phase diamond, the $sp^2$-$sp^3$ mix phase is referred to as the $sp^2$-$sp^3$ composite phase.

In this work, we produced a CPD via a straightforward and cost-efficient approach involving the heat-treatment diamonds under atmospheric pressure conditions. This process resulted in the successful preparation of a negative temperature coefficient diamond with a highly fitted resistance-temperature (R-T) curve. The lowest temperature measurement limit of the CPD was 1 mK. Furthermore, the CPD demonstrated remarkably low sensitivity to magnetic fields and excellent thermal stability, which significantly expands its potential applications as a temperature measurement material. This composite phase structural design was successfully implemented across multiple scales; for example, a micrometer-scale CPD probe and a centimeter-scale 3D printing CPD were developed. The CPD is a promising material for the manufacture of next-generation cryogenic temperature sensors and opens up new potential for functional diamond applications.

## Results

### Structural characterization of CPD

The CPD is fundamentally a result of diamond graphitization. Diamond graphitization is generally considered undesirable, as it impairs the mechanical properties and thermal stability of the diamond[24,30]. Graphitized diamond generally exhibits distinct boundaries[31,32], which can enhance the electrical properties of diamond. However, the occurrence of such boundaries can also deteriorate mechanical properties and thermal stability owing to the presence of significant differences in material properties between the neighboring regions. The synthesized CPD exhibited a different structure from the traditional graphitized diamond. As depicted by the high-resolution transmission electron microscopy (HRTEM) image (Fig. 1a), the island-like amorphous carbon (yellow shadow domains) accompanied by graphite fragments (red shadow domains) was relatively homogeneously embedded in the diamond matrix. Both the amorphous carbon and graphite fragments constitute the $sp^2$-hybridized carbon phase. The diamond matrix phase (blue shadow domains) exhibited an interatomic spacing of 0.206 nm, corresponding to the {111} diamond spots in the fast Fourier transform (FFT) image (Fig. 1a, inset). The existence of this $sp^2/sp^3$ composite phase structure was also verified through other characterizations, including X-ray diffraction (XRD), X-ray photoelectron spectroscopy (XPS), and Raman spectroscopy (Supplementary Fig. 1). The amorphous carbon zone with an average size of 3.55 nm (26% volume fraction) exhibited an interatomic spacing of approximately 0.338−0.357 nm. The graphite fragments zone with a size of 3.01 nm (17% volume fraction) showed an interatomic spacing of 0.338 nm, which corresponded to the graphite fragments {0002} at the arced position of the FFT image with a brighter contrast (Fig. 1a inset). It should be noted that the thickness of the TEM sample may result in inaccuracies on the analysis in the delineation of the amorphous carbon and graphite fragments. Therefore, these values are only provided as a rough reference for the distribution trends of the amorphous carbon zone and graphite fragments zone. The transformation process of diamond to graphite has been controversial. A valuable reference has been provided by detailed studies[25,33–36] on the phase transformation process between graphite and diamond. The higher-magnification

image (Fig. 1b) of the selected dashed zone in Fig. 1a also indicates the possibility of this phase transition process. The (_11_1) diamond plane oriented to the [_112] zone axis was directly transformed into the rectangular-like {0002} graphite plane oriented to the <1010> zone axis (Fig. 1b, red shadow domain 1) and subsequently into the fingerprint-like {0002} graphite plane (Fig. 1b, red shadow domain 2). This transformation process can be represented as d{111} →g{0002}. This transformation process is similar to that of graphite to diamond[33,34]. In addition, the intermediate amorphous carbon phase (Fig. 1b, yellow shadow domain) may be formed during the phase transition process[25,35]. The amorphous carbon also continued to transform into graphite fragments to reach a more stable state (Fig. 1b, red shadow domains). The selected-area electron-diffraction (SAED) image revealed a broad, amorphous carbon halo and face-centered cubic diamond crystalline spots oriented to the [_112] zone axis (Fig. 1c). In contrast, the graphite phase was not prominent, because of its low concentration and the overlapping of its {0002} planes with the amorphous halo. The lower-magnification HRTEM images of different positions (Supplementary Fig. 2a-c) confirmed that the nanocomposite phase structure exhibited a repeatable intermittent periodic distribution in at least a 2 μm-sized region throughout the sample. Furthermore, the phase transition process also promoted the generation of homogeneous and alternating tension and compression stress fields (Supplementary Fig. 2d). The presence of the stress field and the nanocomposite phase structure was also supported by the shoulders observed near the (111) and (220) peaks in the XRD pattern (Supplementary Fig. 1a). The composite phase structure and the

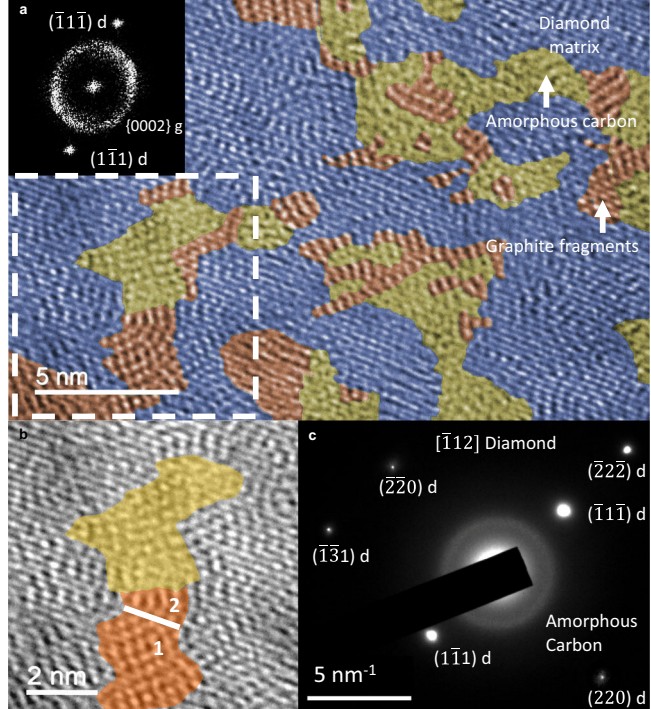

**Fig. 1 | HRTEM images with corresponding FFT and SAED image showing the nanostructure characteristics of the nano composite phase structure. a** (_11_1) diamond matrix (blue shadow domains) embedded by {0002} graphite fragments (red shadow domains) accompanied by amorphous carbon (yellow shadow domains). Inset: FFT pattern showing the crystalline spots embedded in the amorphous halo: a joint broad amorphous halo and arced brighter-contrast {0002} graphite fragments combined with (_11_1) and (1_11) diamond crystalline spots. **b** a magnified HRTEM image corresponding to the dashed box area of (**a**). Red shadow domain 1, red shadow domain 2 and yellow shadow domain: rectangular-like {0002} graphite fragments, fingerprint-like {0002} graphite fragments and amorphous carbon, respectively. **c** SAED pattern corresponding to a.

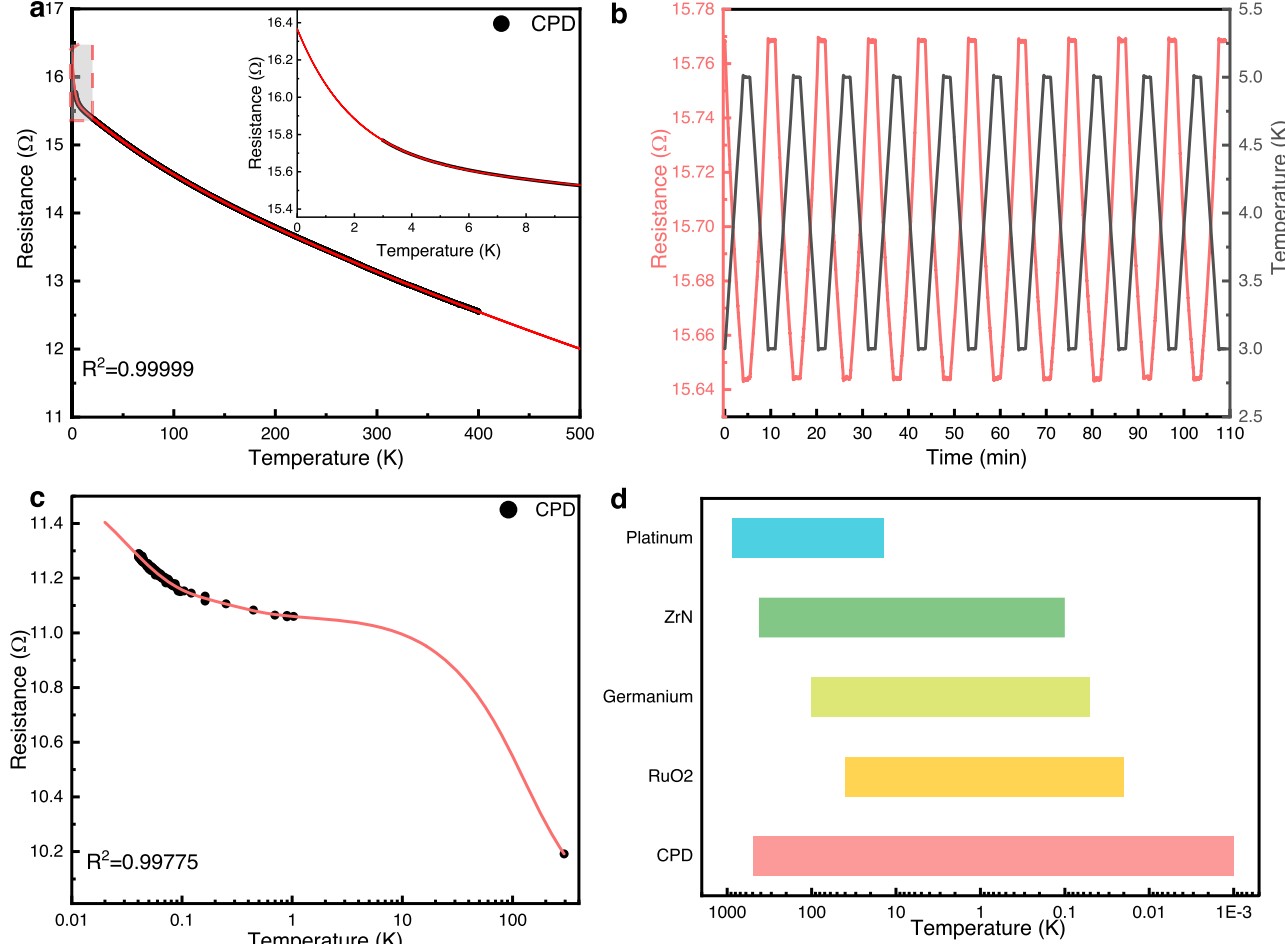

**Fig. 2 | Performance of the temperature sensing characteristics of composite phase diamond (CPD). a** Resistance vs. temperature ($R$-$T$) curve from 3 to 400 K of the CPD sample with initial resistance of 13.13 Ω (room temperature); black line: experimental data; red line: data fitted using the Expdec3 function. Inset: a magnification of the dashed box area. **b** Curve of the resistance response of the CPD during a dynamic response test conducted at 3–5 K. **c** Resistance changes of CPD below 1 K; black dots: experimental data; red line: data fitted using the Expdec3 function. **d** Comparison of the temperature measurement range of the CPD with those of other cryogenic thermometers. Specific values and data sources are listed in Supplementary Table 2. Source data are provided as a Source Data file.

corresponding stress fields could be the primary factor responsible for the enhancement of the diamond high-temperature oxidation resistance[24,37,38].

## Conductivity and temperature coefficient of CPD

The conductivity of the CPD plays a crucial role in its application as a thermistor or temperature-sensing material. High conductivity is essential for ensuring proper operation of the sensor at low temperatures while minimizing self-heating. The CPD showed an exceptional conductivity of 1.2 S·cm⁻¹ at room temperature, which is comparable to that of doping diamond (Supplementary Fig. 3). This is due to the large number of $sp^2$-hybridized amorphous carbon and graphite fragments distributed throughout the diamond matrix. These $sp^2$ carbon with delocalized π-electron system, increased the intrinsically electron transport of the diamond[33,34]. Moreover, the amorphous carbon and graphite fragments coexistence phases randomly and uniformly distributed within the $sp^3$ carbon phase. This can further increase the conductivity of the sample by creating percolation pathways and forming a conductive network[25,35]. In addition, the interfaces between $sp^2$/$sp^3$ phases exist a certain number of defects. These structural irregularities can act as electron scattering centers, contributing to electron mobility and transport enhancement[39]. These constitutes the mechanical basis of the room-temperature electrical conductivity in CPD. Furthermore, the band gap and ionization energy

of CPD were characterized by UV-Vis Diffuse Reflectance Spectroscopy (UV-Vis DRS) (Supplementary Fig. 4a), ultraviolet photoemission spectroscopy (UPS) (Supplementary Fig. 4b), and Photoluminescence spectra (PL) (Supplementary Fig. 4c). The band gap of the CPD is calculated to be about 1.87 eV and 1.80 eV by UV-Vis DRS and PL, respectively. The ionization energy of the CPD is obtained to be 7.84 eV by UPS. All these results are lower than the literature values for diamond[40] (band gap value of 5.47 eV and ionization energy of 81 eV), which is also consistent with the above theory. Figure 2a presents the $R$-$T$ curve of a CPD sample with an initial resistance of 13.13179 Ω (room temperature). The black curve displays the resistance data points acquired at a cooling rate of 2 K·min⁻¹. The curve exhibited a monotonically increasing characteristic throughout the entire test temperature range, signifying a negative temperature coefficient (NTC). Usually, at temperatures close to absolute zero, the resistivity of NTC materials increases sharply. Although this characteristic can enhance the sensitivity of the thermometer, it also restricts the thermometer from measuring lower temperatures. In contrast, the CPD does not encounter such issues. The synthesized CPD showed an almost linear increase in resistance with decreasing temperature from ~400 to 10 K, and below 10 K, the resistance markedly increased. Nevertheless, even at a temperature as low as 3 K, the CPD maintained a very low resistance value, an unusual characteristic of semiconductor materials. This property guarantees that the CPD would not experience failure due to

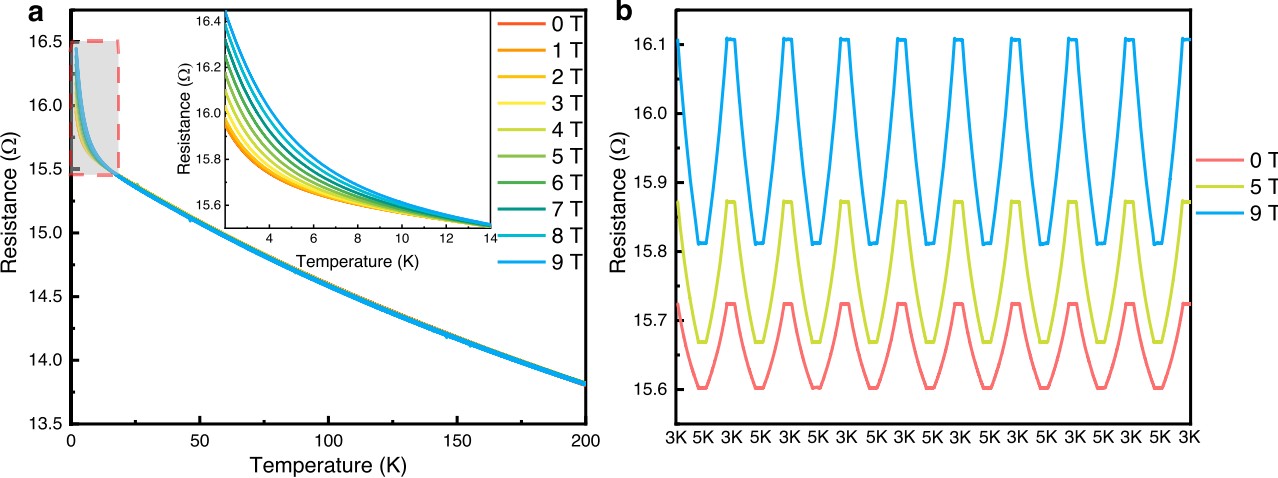

**Fig. 3 | Performance of the temperature sensing characteristics of the CPD. a** Resistance shifts of the CPD under different applied magnetic fields. Inset: 2–14 K zoomed range. **b** Dynamic response test results under different applied magnetic fields. Source data are provided as a Source Data file.

insulation. The data were fitted using a three-phase exponential decay function (Expdec3) and extrapolated to the temperature range of 0 - 500 K (indicated by the red curve). The coefficient of determination $R^2$ for the fitted curve was 0.99999, verifying the goodness of fit model. Furthermore, a dilution refrigerator was used to characterize the resistance changes of another CPD sample below 1 K. The result is shown in Fig. 2c and Supplementary Fig. 5a. The resistance of the CPD also remains monotonically increasing within the 40 mK-1 K temperature interval, with no inflection points or abrupt changes. In addition, the $R$-$T$ curve of the CPD below 1 K has a similarly high $R^2$ of 0.9978 when fitted using the Expdec3 function (Fig. 2c). The above results show that the physical properties of the CPD remain stable when it approaches absolute zero from room temperature. Moreover, the regular residuals, defined as the difference between the predicted and experimental resistance values, were primarily in the range of ±0.008 arb. units (Supplementary Fig. 6a), which suggests that the theoretical calibration accuracy of the CPD could reach as high as 8 mK under a temperature range of 3 to 400 K. When only the data corresponding to temperatures below 10 K were fitted, the value of the regular residuals was less than ±0.001 arb. units (Supplementary Fig. 6b and 6c). Therefore, the CPD can achieve a temperature measurement resolution of 1 mK at <10 K, which is vital for the practical implementation of cryogenic temperature sensors. In the dynamic response test (Fig. 2b), the resistance of the CPD exhibited an excellent correspondence to the temperature. Furthermore, we collected data in the range of 2.999−3.001 K (Supplementary Fig. 6d). The variance and standard deviation of 842 data points were $5.62547 \times 10^{-8}$ and $2.37181 \times 10^{-4}$ respectively, signifying a highly concentrated data distribution. These data were further divided into two intervals: 2.999 ≤ T < 3 K and 3 ≤ T < 3.001 K. The calculated intervals with 95% confidence for the former and latter were 15.76844−15.76849 Ω and 15.76837−15.76841 Ω, respectively (Supplementary Table 1). These two separate confidence intervals confirm that the temperature measurement accuracy of the CPD at 3 K can reach 1 mK. According to the fitted results obtained using the Expdec3 function, the resistance of the sample at 2 mK was computed as 16.36614 Ω, and the resistance at 1 mK was 16.36652 Ω. The difference between these two predicted values is larger than that between the resistance values in the temperature intervals of 2.999 - 3 K and 3 - 3.001 K. This proved that the theoretical detection temperature of the CPD can reach 1 mK, a level not yet achieved by any existing cryogenic temperature sensor. Supplementary Fig. 5b displays the resistance response over time with a temperature increase from 3 to 400 K. It demonstrates the excellent stability of the CPD during temperature measurements. Figure 2d

compares the temperature measurement ranges of the CPD with those of other cryogenic thermometers. The CPD demonstrated significant advantages as it could measuring temperatures at the mK level. In addition, as shown in Supplementary Fig. 7, put the CPD into liquid nitrogen from room temperature until stable, the corresponding response time is 3.48 s. The time taken to respond to 90% of a step change in temperature ($T_{0.9}$) is only 2.04 s. Such a capability can facilitate the translation of quantum computing, quantum simulation, and related technologies from research to practical applications. Furthermore, the CPD was successfully synthesized at various scales, for examples, a micrometer-scale CPD probe (Ø=1 μm) fabricating by focused ion beam (FIB), a millimeter-scale bulk CPD and a complex structure with CPD via 3D printing (Supplementary Fig. 8). These samples of different sizes can be applied to different occasions, enhancing the potential application value of CPD. Supplementary Fig. 9 further illustrates the repeatability of CPD samples with different sizes.

## Magnetoresistance of CPD
Cryogenic temperature measurements are considerably limited by the presence of magnetic fields. These fields induce reversible calibration shifts, leading to inaccurate temperature readings. These shifts are temporary, and the sensors revert to their zero-field calibration upon the removal of the magnetic field. The sensitivity of the sensor to magnetic fields considerably influences its application value. Figure 3a, Fig. 3b, and Supplementary Fig. 10 depict the resistance changes of the CPD in a magnetic field environment. Under temperature conditions of >14 K, the CPD was almost insensitive to magnetic field changes (Fig. 3a, Supplementary Fig. 10). Even at <14 K, the magnetoresistance changes of the CPD remained relatively small. At a temperature of 2 K and under a strong magnetic field of 9 T, the resistance shifts rate of the CPD was only ~3% (Fig. 3a inset). The insensitivity of the CPD to magnetic fields is significant in applications such as nuclear magnetic resonance. In addition, the CPD displayed linear magnetoresistance properties under magnetic fields (Supplementary Fig. 10). This characteristic can simplify the calibration process for temperature sensors in magnetic field environments and endows the CPD with potential applications in novel quantum linear magnetoresistive devices, particularly magnetic sensors[4].

## Thermal stability of CPD
Cryogenic temperature sensors are also operated and stored under room temperature or high temperature conditions. The resistance of the temperature sensor may undergo shifts following the

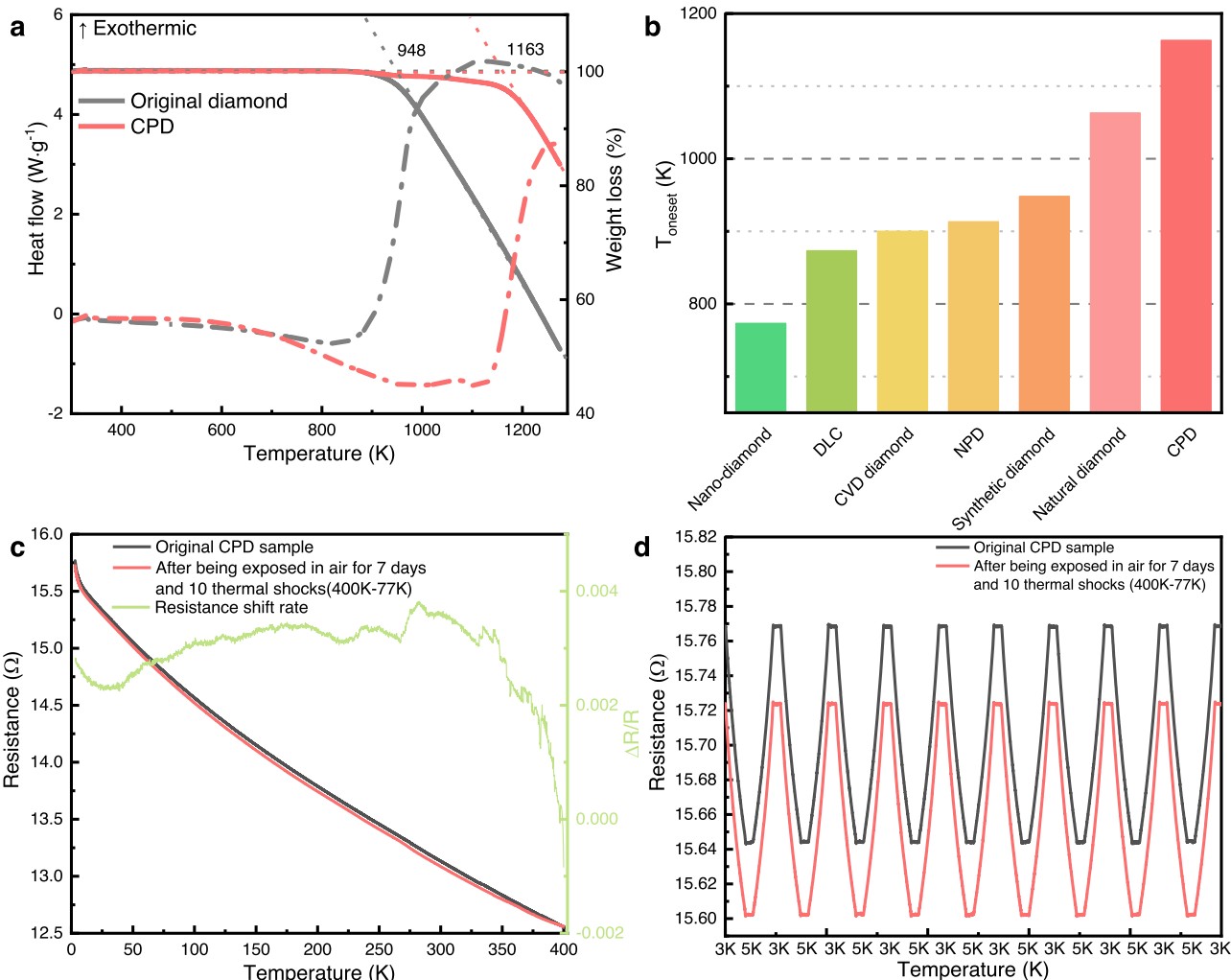

**Fig. 4 | Thermal stability of the CPD. a** Thermogravimetric and differential scanning calorimetry curves of the CPD, measured in air at a heating rate of 5 K·min⁻¹. Solid lines: TG curves, refer to right y-axes. Dash dot lines: DSC curves, refer to left y-axes. **b** Comparison of the CPD with other diamond and diamond-like materials in terms of thermal stability[49–52]. The data for the synthetic diamond are represented by the data for the original diamond. **c** Resistance shifts of the CPD, measured after exposure to air for seven days and 10 thermal shocks at 400 to 77 K. **d** Dynamic response test of the CPD at 3–5 K after thermal shocks. Source data are provided as a Source Data file.

measurement of a large temperature difference. This significantly challenge the thermal stability of the cryogenic temperature sensor, particularly for carbon-based sensors. We investigated the intrinsic thermal stability of the CPD (Fig. 4a) and its resistance variation after thermal cycling (Fig. 4c, d). The oxidation of diamonds in air typically encompasses two processes: direct diamond oxidation and diamond oxidation after graphitization[41]. Graphite and other $sp^2$-hybridized carbon materials typically exhibit a higher susceptibility to oxidation in air than $sp^3$-hybridized diamonds[42,43]. Consequently, the research on enhancing the oxidation resistance of diamonds often centers on diamond graphitization-prevention strategies[24,44]. Our study found that the incorporation of an $sp^2$-hybridized carbon phase within the diamond structure did not reduce the onset oxidation temperature. Instead, the presence of the $sp^2$-hybridized carbon phase significantly increased onset oxidation temperature of the CPD: the onset oxidation temperatures of the original diamond and the CPD were 948 K and 1163 K, respectively (Fig. 4a). The CPD exhibited greater thermal stability than other diamond materials (Fig. 4b). This finding contradicts the commonly held belief[42] that the presence of $sp^2$-hybridized carbon bonds reduces thermal stability. We suppose that these anomalous characteristics may be related to the structure of CPD. Compared to traditional graphitized diamond, the microstructural characteristics of

CPD is $sp^3$ carbon dominated (å 50% $sp^3$ carbon) with $sp^2$ nano carbon phase uniformly discrete distributed. This microstructure avoids the existence of larger continuous $sp^2$ carbon bonds and protects $sp^2$ nano carbon phase from direct contact with oxygen, which can inhibit the $sp^2$ carbon clusters increasing at high temperatures[45]. This study is the first instance in which the high-temperature oxidation resistance of diamond is enhanced without the application of high-pressure treatments, which were adopted in previous research[24,46]. Supplementary Fig. 11 illustrates the impacts of different heat treatment conditions on the thermal stability of the CPD. With increasing heat treatment duration, the thermal stability of the CPD was significantly enhanced, while the increase in the heat treatment temperature from 1250 to 1550 °C negligibly influenced the thermal stability of the CPD. Currently, no definitive explanation exists for this unusual phenomenon. We consider that the uniform distribution of the $sp^2$-hybridized carbon phase within the diamond matrix (Fig. 1a, Supplementary Fig. 2b, c), increased the number of lattice distortions and the stress field in the CPD. Moreover, the $sp^2$ carbon phase has a greater thermal expansion coefficient than that of diamond[47]. The thermal expansion of those $sp^2$ nano carbon phases further increase the internal stresses of CPD, which is the main factor to enhance the diamond's resistance to high-temperature oxidation resistance performance. The SEM images

shows the differences in the oxidation processes between diamond and CPD (Supplementary Figs. 12 and 13) more visually. The presence of these lattice distortions and the stress field augments the nanoscale properties of the diamond, counteracts the graphitization-induced reduction in strength, and inhibits the graphitization and oxidation processes of the diamond in air[24].

The significant improvement in the high-temperature oxidation resistance demonstrates the excellent thermal stability of the CPD. Figure 4c compares the resistance change between the original CPD sample and the sample exposed to air for seven days without any protection and then subjected to 10 thermal shocks (77–400 K). Although the overall resistance of the sample increased slightly after thermal shocks, the change rates was <0.4%. These results demonstrate that the CPD exhibited exceptional electrical resistance stability in response to temperature changes, consistent with the dynamic response test results (Fig. 4d). After thermal shocks, the R-T trend of the CPD remained unchanged.

## Discussion

In summary, we successfully synthesized a CPD with remarkable electrical properties and thermal stability through a straightforward approach. The CPD demonstrated its capability and stability as a temperature-sensing material in extreme environments, verifying our original design intention. The lowest temperature measurement limit of the CPD reached 1 mK, which can contribute to the advancement of low-temperature physics and quantum physics. In addition, the composite phase structure considerably influenced the diamond properties. This nanoscale crystal reconstruction allowed for the transformation of the diamond from a superhard material to a functional material. The successful synthesis of the CPD will create new paths towards diamond applications in precision measurement, quantum computing, medical devices, and space technology.

## Methods

### Sample synthesis

The original diamond is a commercially available synthetic diamond powder provided by Tianjian Carbon Material Co., Ltd., China, and belongs to the Type 1b diamond. The diamond particles (80 μm) were fully mixed with acrylic acid ammonium salt polymer (≥99%), acrylamide (≥99%), N, N'-Methylenebisacrylamide (≥99%), 2-hydroxy-2-methylpropiophenone (≥99%), diphenyl(2,4,6-trimethyl benzoyl) phosphine oxide (≥99%) and water in a specific weight ratio of 60:3.3:2.4:4:0.24:7.3. 3D printing was carried out using a DIW 3D printer (3D Bio-Architect® Work Station) from Regenovo Biotechnology Co., Ltd., China. The printing was carried out with a line width of 0.4 mm, utilizing a closely spaced pattern that can be achieved through a crosshatch printing method. During the printing process, preliminary curing was accomplished by exposure to a 365 nm ultraviolet lamp. Then, the sample was dried at 80 °C in an oven (Froilabo, France) for 4 hours. Subsequently, it was sintered in an Ar atmosphere at 1250 °C and atmospheric pressure to obtain the CPD. All samples were ultrasonically cleaned in anhydrous ethanol. The sintering process is conducted in a tube furnace (BTF-1700C, ANHUI BEQ EQUIPMENT TECHNOLOGY CO., LTD., China). Unless otherwise specified, the sintering time for the samples presented in the manuscript is 1800 min.

### Structure and phase characterization

The microstructures of the CPD were investigated via HRTEM and SAED using a transmission electron microscope (JEM-2011F, JEOL Ltd., Japan) with an accelerating voltage of 200 kV. To depict the distribution of the amorphous carbon zone and graphite fragments zone in CPD, we performed pseudo-color calibration of the phase regions of multiple HRTEM images. Then ImageJ was employed to quantitatively analyze the corresponding areas, ultimately determining the content of each phase. The stress field of the nanocomposite phase structure was characterized via geometric phase analysis using the background-filtered image in Strain++ software. The component phases and structure of the CPD were characterized via TEM, XRD (Smartlab, Rigaku, Japan) with CuK$_\alpha$ radiation (I = 1.541838 Å), XPS (PHI 5802, PHI, USA), and Raman spectroscopy (alpha300 R, WITec, Germany) with an excitation wavelength of 532 nm. Prior to XPS measurement, a 0.5 nm-thick Au coating was sputtered onto the samples to calibrate all spectra by shifting the Au 4f-7 peak to 84.0 eV.

### Property measurements

The conductivity of the sample was measured using a four-point probe meter (CXT2665, Changzhou Xinyang Electronic Technology Co., Ltd., China), with a distance of 1.59 mm between the probes. The UV-Vis DRS was measured by PerkinElmer Lambda950. The band gap of CPD was derived from the absorption spectra using the Tauc plot method. The UPS was measured by Escalab Xi+. A helium gas discharge lamp emitting light in the ultraviolet region (He I, E$_{hv}$ = 21.22 eV) was used to measure the valence band spectra. The PL spectra of CPD were measured by Edinburgh FLS1000, with a laser wavelength of 532 nm. The resistance values of the CPD at different temperatures and magnetic fields were measured using a physical property measurement system (Quantum Design PPMS® DynaCool™, Quantum Design Inc., USA) equipped with a Cernox™ temperature sensor. The high-temperature oxidation resistance of the CPD was studied using a thermal analysis system (TGA/DSC 3+, Mettler Toledo, Switzerland) in air with a heating rate of 5 K min$^{-1}$. The measurements were conducted in air over a temperature range of 20–1000 °C.

### Reporting summary

Further information on research design is available in the Nature Portfolio Reporting Summary linked to this article.

## Data availability

Source data are provided with this paper and can be found at reference[48]. Source data are provided with this paper.

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

## Acknowledgements

J.L. acknowledges the Shenzhen-Hong Kong Science and Technology Innovation Cooperation Zone Shenzhen Park Project (Project No: HZQB-KCZYB-2020030), the Hong Kong General Research Fund (GRF) Scheme (Project No: CityU 11216219), the Research Grants Council of Hong Kong (Project No: AoE/M–402/20.) and the Hong Kong Innovation and Technology Commission via the Hong Kong Branch of National Precious Metals Material Engineering Research Center. We thank Tianjian Carbon Material Co., Ltd. for providing diamonds. The authors would like to acknowledge Prof. Danfeng LI, Prof. Kun YU and Mr. Tit Wah CHAN for their consistent encouragement and support. The authors would like to acknowledge Ms. Ruonan ZHAO, Mr. Zhicheng PEI, Dr. Bingqiang WEI and Mr. Jingzhuo ZHOU for fruitful discussions.

## Author contributions

J.L. proposed and supervised the project; J.L. and J.A.Y. designed the experiments. J.A.Y. and M.L.M. synthesized the CPD and measured electrical properties. Y.Y. performed the TEM observations and FIB experiments. J.Y.T. measured the resistance of CPD at temperatures below 1 K. J.L.J., H.L., and Y.H.C performed the structure characterization. Y.X.C, F.C.L, B.B.Z, Z.Y.M., L.W. and Y.H.H measured the properties. J.A.Y. and Y.Y. analyzed the data and wrote the manuscript according to the contributions of all authors.

## Competing interests

The authors declare no competing interests.
