## [Peer Review File · Nature Communications]

Diamond with Sp^2 - Sp^3 Composite Phase for Thermometry at Millikelvin TemperaturesREVIEWER COMMENTS

Reviewer #1 (Remarks to the Author):

The development of low-temperature physics will improve the understanding of the universe, matter, and technology. For low-temperature physics, lowering the temperature measurement limit as much as possible is the eternal pursuit of scientists in this field. This work reports the achievement of the lowest temperature measurement limit to date by developing an amorphous carbon/graphite/diamond composite. The topic is interesting and the results are stirring. Thus, I agree with the publication of this paper. But several key issues needed to be clarified before the paper can be fully accepted for publication.

Major:

(1) The term "dual-phase diamond" is inappropriate. Diamond itself is the name of a phase, just like the α or γ phase in metal. Furthermore, for the materials scientists in ceramics, the term "diamond" refers to the cubic diamond. Besides cubic diamond, the existence of hexagonal diamond is theoretically predicted. Therefore, the term "dual-phase diamond" will mislead readers into that it is a composite of cubic diamond and hexagonal diamond. However, island-like amorphous carbon accompanied by graphite fragments embedded in the diamond matrix is the main structural feature of the samples prepared in this paper. Thus, "carbon/carbon composite" or other similar terms may be more appropriate.

(2) The band gap of the diamond is 5.47 eV so it does not conduct electricity. In this work, the authors show a distinct structure in which the diamond matrix embedded the island-like amorphous carbon and graphite fragments. At the same time, this structure shows an exceptional conductivity of 1.2 S/cm at room temperature. It is difficult to understand that such appreciable conductivity can be achieved in a diamond matrix. In addition, some studies (Nature, 2022, 607, 486; Nature Materials, 2023, 22, 42; National Science Review, 2022, 9, nwab140; Cell Reports Physical Science, 2021, 2, 100575) also achieved an outstanding electrical property. A detailed comparison with these studies in terms of structure and electrical properties is necessary to understand the unique role of the structure designed in this work.

(3) The mechanism of diamond graphitization is an ancient and still controversial academic issue. In this work, the transformation process was determined only by HRTEM. The evidence is weak and the detailed phase transformation is lacking. In fact, the phase transformation of diamond is more complex due to the existence of amorphous carbon. For example, which is formed first, amorphous carbon or graphite? How are they formed? Does the formation of amorphous and graphite depend on a specific orientation relationship? This issue is really challenging. If it is difficult to give sufficient evidence for the mechanism of diamond graphitization, a discussion can be supplemented by referring to the mechanism from graphite to diamond (Nature, 2022, 607, 486) or from amorphous carbon to diamond (Nature Materials, 2023, 22, 42).

(4) Temperature resolution and measurement limit are two key parameters for temperature measurement. In addition, for rapid cooling and heating applications, the time resolution of temperature measurement is also extremely important. For this aspect, it may be necessary to supply some data or discussion.

(5) As the authors mentioned, the research on enhancing the oxidation resistance of diamond often centers on diamond graphitization-prevention strategies. Interestingly, this work found that the incorporation of an sp^2 -hybridized carbon phase within the diamond structure increases the onset oxidation temperature. An explanation should be given.

(6) What is the sample size used in Figs. 2 to 4? In other words, does the sample size affect the performance of the temperature-sensing characteristics? The details of sample synthesis, such as the sintering method, equipment model, and sintering pressure, should be given. The authors also should provide detailed descriptions of the 3D-printed current structures in the Method section.

Minor:

(1) In the introduction, the authors said that "(Line 48) However, compared with diamond, sp^2 -hybridized carbon materials are fragile and unstable." This statement is inappropriate. Graphite is a typical sp^2 -hybridized carbon material, and it is more stable than diamond at room temperature and standard atmospheric pressure.

(2) In this work, the amorphous carbon has an average size of 3.55 nm and the graphite fragment has an average size of 3.01 nm. What is the thickness of the TEM sample in Fig. 1? Generally, the thickness of the TEM sample is about tens to hundreds of nanometers. If so, a misjudgment of

their sizes could result from the overlap of the amorphous carbon and graphite.

(3) The calculation details of the phase content should be given.

(4) The authors' previous works and numerous previous studies (such as Matter 2023, 6, 7) have shown us that designing a dual-phase structure is a powerful approach to improving the mechanical properties of materials. However, it is difficult to imagine from these works that a dual-phase material can overcome the bottleneck of low-temperature measurement. Therefore, a related description should be supplied in the introduction.

(5) I can't see the specific text in the inset table in Fig. 2.

(6) Although the details of the strength test are described in the method, the relevant data are not given in the manuscript.

Reviewer #2 (Remarks to the Author):

The authors successfully synthesized a DPD with remarkable electrical properties and thermal stability through a straightforward approach. The DPD seems to demonstrate its capability and stability as a temperature sensing material in extreme environments, implying the potential application to the low-temperature physics and quantum physics. In my opinion, most of the contents of the manuscript are well organized and easy to read in different fields. In particular, this work may extend the application of diamond from a superhard material to a functional material. However, some critical issues prevent the paper from being published in its current version.

1. The authors made a strong claim that the lowest temperature measurement limit (in fact, it is a prediction) of the DPD was 1 mK based on the true low temperature measurement down to 2K. Meanwhile, in the introduction, the authors mentioned that "traditional thermometers can only measure temperatures higher than 20 mK". So I asked the authors to do additional experiments at the current limit (e.g., about 20 mK or slightly higher) and repeated the measurements by using several DPD samples (to exclude the effect of size and quality of samples).

2. In the published paper [Chem. Mater. 33, 8722 (2021); <https://doi.org/10.1021/acs.chemmater.1c02683>], the temperature-dependent resistivities of Ti₂Co shows the semiconducting behavior from the temperature range of 300 K to 2 K [see Fig 4c and 4d]. Strikingly, it shows superconductivity at ~0.75 K. In this case, some materials exhibit abnormal transition at a very low temperature beyond what we usually expected. From this point, it is a premature conclusion that the lowest temperature measurement limit of DPD was 1 mK without the solid experimental evidence (only by using an extrapolation method).

3. If the authors could not provide a further/new experimental evidence in the revision, I suggest the author to revise their conclusion to "the predicted temperature measurement limit of the DPD reaches the mK level" and transfer to another journal.

Minor questions :

4. How did the authors determine the exact temperature value and collect the data in a range of 2 mK as shown in Extended Data Fig. 4d? Furthermore, what standard was used to divide the data into two intervals?

5. From the data shown in the paper, the resistance values are consistent at the same temperature. Are these data obtained from the only one sample? Would the measured values change with the quality and size of the DPD samples?

6. Could the authors theoretically explain how the DPD works as a cryogenic temperature sensor to some extent beside using a three-phase exponential decay function.

Reviewer #3 (Remarks to the Author):

This paper introduces a dual-phase diamond (DPD) as a potential thermometer to probe temperatures below millikelvin levels. The authors synthesized DPD through the sintering of diamond powders with specific chemicals. Utilizing various analytical techniques, they confirmed the presence of graphitic phases (along with amorphous transition phases) within the diamond structures, resulting in the formation of DPD. The temperature dependence of the resistance of the produced DPD was measured, revealing a negative temperature coefficient trend. Extrapolating this dependence led them to conclude that the theoretical detection temperature of DPD could reach 1 mK, a level hitherto unattained by existing cryogenic temperature sensors.

While I find the observed temperature range and magnetic susceptibility of DPD intriguing, I must emphasize that the experimental methods and data analysis demand a more rigorous treatment for this work to merit the aforementioned statement, as described below.

Firstly, the current dataset does not validate the operation and usefulness of thermometry at millikelvin temperatures. The data fitting was restricted to the range of 3-400 K, and all discussions related to the millikelvin thermometer operation rely on extrapolating the fitting curve to temperatures below 3 K. For instance, a study involving RuO₂-based thermometers for millikelvin temperatures (Cryogenics (Guildf). 32, 1167 (1992)), also cited as Ref. 32, experimentally demonstrated the R-T dependence down to 50 mK, which is two orders of magnitude lower than the lower limit experimentally confirmed in the present manuscript. Significantly, the data in Ref. 32 displays a plateau in the R-T curve below 0.1 K. This deviation from the conventional exponential decay dependence strongly implies the necessity of experimental verification of thermometry within the claimed temperature range. The exponential decay tendency of the present DPD cannot be assumed to behave straightforwardly.

Secondly, the details regarding sample preparation and measurement methods are rather unclear. The methodology for sample preparation lacks several key points. For example, the type of diamonds used (Type-1b, 2a), information about impurity concentrations, whether HPHT or CVD diamonds were used, etc., remain unspecified. Furthermore, the phrase "mixed in a specific ratio" with chemicals (Line 274-281) requires clarification. In the measurement section, critical information such as the size of the measured sample (which could affect resistance measurements) and the methodology employed for making electrical contacts in the R-T data are absent. Is a four-point probe used even for low-temperature measurements? Additionally, it is mentioned that they fabricated micron-sized structures using FIB; however, it is important to note that the mere fabrication of micron-sized structures does not necessarily imply that these structures of DPD can effectively function as thermometers. These technical aspects are crucial for enabling the reproducibility of data by other researchers in the future. It is also noteworthy that the present paper only reports data from a single sample for R-T analysis. Thus, questions arise about reproducibility and potential variations in device fabrication.

Lastly, from a conceptual standpoint, I find myself in disagreement with the authors' motivation for temperature measurement. The manuscript refers to "traditional thermometers being limited to temperatures above 20 mK" based on Ref. 32. However, this limitation does not stem from the

thermometer materials employed in Ref. 32. Instead, it arises from the technical challenges associated with the dilution refrigerator used in Ref. 32. Although dilution refrigerators can achieve temperatures in the millikelvin range, reaching this level is not easy. Moreover, the abstract states, "Currently, millikelvin temperatures and below are measured through the characterization of a certain thermal state of the system as there is no traditional thermometer capable of measuring temperatures at such low levels." Yet, the approach of characterizing a certain thermal state of the system (presumably indicating a quantum gas) for temperature measurement is entirely reasonable. This technique circumvents the heat capacity issue, wherein the thermometer's heat capacity could influence temperature measurements for samples with small heat capacities or at extremely low temperatures. On this note, I sense that the present paper might not have thoroughly considered the practical implementation of this "thermometer material" as an actual thermometer.

Given my reservations concerning the quality of the data and analysis, I hold significant concerns regarding the suitability of publishing this work in Nature Communications, a journal that necessitates broad interest.

RESPONSE TO REVIEWERS' COMMENTS

We appreciate the referees for providing valuable comments. The responses are printed in blue, and highlight changes made in the revised version of our manuscript in red for easy reference.

Reviewer #1:

The development of low-temperature physics will improve the understanding of the universe, matter, and technology. For low-temperature physics, lowering the temperature measurement limit as much as possible is the eternal pursuit of scientists in this field. This work reports the achievement of the lowest temperature measurement limit to date by developing an amorphous carbon/graphite/diamond composite. The topic is interesting and the results are stirring. Thus, I agree with the publication of this paper. But several key issues needed to be clarified before the paper can be fully accepted for publication.

Response: We really appreciate that the reviewer thinks our work could be accepted. Thank you for your valuable comments and feedback on our scientific paper. We greatly appreciate your insightful observations and are committed to addressing them effectively. We have tried our best to address the raised questions one by one in the response below, and revised our manuscript accordingly.

Major:

(1) The term “dual-phase diamond” is inappropriate. Diamond itself is the name of a phase, just like the α or γ phase in metal. Furthermore, for the materials scientists in ceramics, the term “diamond” refers to the cubic diamond. Besides cubic diamond, the existence of hexagonal diamond is theoretically predicted. Therefore, the term “dual-phase diamond” will mislead readers into that it is a composite of cubic diamond and hexagonal diamond. However, island-like amorphous carbon accompanied by graphite fragments embedded in the diamond matrix is the main structural feature of the samples prepared in this paper. Thus, “carbon/carbon composite” or other similar terms may be more appropriate.

Response: We thank the reviewer for the constructive suggestion. I see the issue with the term "dual-phase diamond." The confusion arises partly due to our oversight in not providing a more detailed definition of this term and partly because of the inherent

complexity of carbon materials. As the reviewer mentioned, cubic and hexagonal phase diamonds do exist. While most general readers might not find it difficult to understand the reference to "dual-phase" in the manuscript, not providing a clear definition in the title could potentially mislead a small portion of professional readers to interpret it as a mixture of cubic and hexagonal phase diamond materials. We appreciate the suggestion to clarify this, and we will make the necessary revisions to avoid any misunderstandings. However, carbon/carbon composite are typically considered as an abbreviation for carbon fiber-reinforced carbon-matrix (C/C) composite materials and are widely used. If "DPD" were to be replaced with "carbon/carbon composite" or similar terms, it might lead to potential confusion in conveying the information within the manuscript. Therefore, we believe that providing a more precise definition of "**Diamond with sp^2 - sp^3 composite phase (CPD)**" would be more appropriate. In this context, the manuscript categorizes all carbon phases containing sp^2 -hybridized carbon (such as amorphous carbon and graphite fragments), into one category, while the sp^3 -hybridized carbon within the diamond matrix is categorized into another. This not only aligns with the content presented in the manuscript but also clearly highlights the focus of this work for the readers' convenience.

(2) The band gap of the diamond is 5.47 eV so it does not conduct electricity. In this work, the authors show a distinct structure in which the diamond matrix embedded the island-like amorphous carbon and graphite fragments. At the same time, this structure shows an exceptional conductivity of 1.2 S/cm at room temperature. It is difficult to understand that such appreciable conductivity can be achieved in a diamond matrix. In addition, some studies (Nature, 2022, 607, 486; Nature Materials, 2023, 22, 42; National Science Review, 2022, 9, nwab140; Cell Reports Physical Science, 2021, 2, 100575) also achieved an outstanding electrical property. A detailed comparison with these studies in terms of structure and electrical properties is necessary to understand the unique role of the structure designed in this work.

Response: We thank the reviewer for the constructive suggestion. Recently, some studies have effectively improved the conductivity of diamond matrix by introducing sp^2 hybridized carbon atoms, and the relevant performance is summarized as Table 1. Luo et.al.¹ found the mixed sp^2 and sp^3 hybridized carbon atoms in Gradia interfaces could increase electrical conductivity without obvious hardness loss. Sp^2 carbon enabled samples to achieve semiconductor characteristics. With higher synthesis temperature, sp^2/sp^3 ratio decreases and electrical resistivity of samples increases. Li

et.al.² prepared the composite composed of nanodiamond and disordered nanomultilayer graphene. The composite shows the excellent electrical conductivity of 670 - 1240 S·m⁻¹ at room temperature. This work demonstrates that the disordered graphene connected with diamond via sp² or sp³ bonds can get a high conductivity. Zhang et.al.^{3,4} systematically studied amorphous carbon materials with different sp²/sp³ bonding fractions obtained from compressed fullerene under 15 and 25 GPa. These amorphous (AM) carbon materials are mainly composed of sp²-disordered nanomultilayer graphene and sp³-dominant dense disordered nano-fragments. The band gap of these AM carbon materials was related to the content of sp²-hybridized carbon. The higher the sp² carbon content, the narrower the band gap and the better the conductivity. Therefore, the introduction of sp²-hybridized carbon into the diamond matrix has been shown to be effective in improving the electrical conductivity of diamond.

We believe that the above theoretical and literature bases are equally applicable to this study. A large number of sp²-hybridized amorphous carbon and graphite fragments are distributed in the diamond matrix. These sp² carbon with delocalized π -electron system, facilitated the intrinsically electron transport of the diamond^{1,2}. Moreover, the coexistence phases of amorphous carbon and graphite fragments randomly and uniformly distributed within the sp³ carbon phase. This can further increase the conductivity of the sample by creating percolation pathways and forming a conductive network^{3,4}. These theories constitutes the mechanical basis of the room-temperature electrical conductivity in CPD. In addition, the interfaces between sp²/sp³ phases exist a certain number of defects. Such structural irregularities can act as electron scattering centers, contributing to enhance electron mobility and transport⁵.

Fig. R1 Spectroscopic analysis of energy band information of CPD. a, UV-Vis spectrum. b, UPS spectrum. c, PL spectrum.

The band gap and ionisation energy of CPD were characterised by UV-Vis Diffuse Reflectance Spectroscopy (UV-Vis DRS) (Fig. R1a), ultraviolet photoemission spectroscopy (UPS) (Fig. R1b) and Photoluminescence spectra (PL) (Fig. R1c). The UV-Vis diffuse reflectance spectroscopy (UV-Vis DRS) plots were processed using the Tauc plot method⁶. An optical band gap of 1.87 eV was obtained for the CPD, which is similar to the value (1.80 eV) obtained from the PL spectrum. Furthermore, the ionisation energy of the CPD was calculated to be 7.84 eV by ultraviolet photoemission spectroscopy (UPS). All these results are lower than the literature values for diamond (band gap value of 5.47 eV and ionisation energy of 80 eV)⁷, which is also consistent with the above theory.

Table R1. Electrical conductivity and band gaps of different sp²/sp³ composite carbon materials

Sample	Electrical conductivity	Band gaps	Ref.
Composite composed of nanodiamond dispersed in disordered multilayer graphene	670-1240 S·m ⁻¹	/	2
AM carbon materials with sp ³ fraction of 65-95%	/	1.5-2.2 eV	3
AM carbon materials with sp ³ fraction of 47-67%	/	0.1-0.3 eV	4
C/C composites	2.0-5.9*10 ⁵ S·m ⁻¹	/	8,9
Glassy Carbon with 0% sp ³ fraction	/	0.01eV	10
ta-C with 80-88% sp ³ fraction	/	2.5eV	
Mixed sp ² -sp ³ carbon allotropes of interpenetrating graphene networks	/	0.48-0.92	11
Zigzag type interpenetrating graphene networks		0.36 a- 0.49 eV	12

We have made the necessary revisions in the manuscript, which now read as follows: The CPD showed an exceptional conductivity of 1.2 S·cm⁻¹ at room temperature, which is comparable to that of doping diamond (Extended Data Fig. 3). This is due to the large number of sp²-hybridized amorphous carbon and graphite fragments distributed throughout the diamond matrix. These sp² carbon with delocalized π -electron system, facilitated the intrinsically electron transport of the diamond^{1,2}. Moreover, the coexistence phases of amorphous carbon and graphite fragments randomly and uniformly distributed within the sp³ carbon phase. This can further increase the conductivity of the sample by creating percolation pathways and forming a conductive network^{3,4}. In addition, the interfaces between sp²/sp³ phases exist a certain number of

defects. These structural irregularities can act as electron scattering centers, contributing to enhance electron mobility and transport⁵. These constitutes the mechanical basis of the room-temperature electrical conductivity in CPD. Furthermore, the band gap and ionization energy of CPD were characterized by UV-Vis Diffuse Reflectance Spectroscopy (UV-Vis DRS) (Fig. R1a), ultraviolet photoemission spectroscopy (UPS) (Fig. R1b) and Photoluminescence spectra (PL) (Fig. R1c). The band gap of the CPD is calculated to be about 1.87 eV and 1.80 eV by UV-Vis DRS and PL, respectively. The ionization energy of the CPD is obtained to be 7.84 eV by UPS. All these results are lower than the literature values for diamond (band gap value of 5.47 eV and ionization energy of 80 eV)⁷, which is also consistent with the above theory.

(3) The mechanism of diamond graphitization is an ancient and still controversial academic issue. In this work, the transformation process was determined only by HRTEM. The evidence is weak and the detailed phase transformation is lacking. In fact, the phase transformation of diamond is more complex due to the existence of amorphous carbon. For example, which is formed first, amorphous carbon or graphite? How are they formed? Does the formation of amorphous and graphite depend on a specific orientation relationship? This issue is really challenging. If it is difficult to give sufficient evidence for the mechanism of diamond graphitization, a discussion can be supplemented by referring to the mechanism from graphite to diamond (Nature, 2022, 607, 486) or from amorphous carbon to diamond (Nature Materials, 2023, 22, 42).

Response: Thank you for the constructive suggestions provided by the reviewer. As the reviewer notes, the intertransformation of the sp^2 and sp^3 carbon phases in carbon materials has various forms. In particular, there are more controversies over how the intertransformation occurs in the two materials: diamond and graphite. In this study we try to combine the mature mechanisms that exist to express our views on this academic issue.

For the transformation process from graphite to diamond, Luo et al.¹ proposed that the growth of diamond is accomplished by advancing the formation (diamond nucleation) and migration (diamond growth) of the interface. The interface between graphite and cubic diamond exhibit the following orientation relations: $[1\bar{2}10]G//[1\bar{1}0]CD$. The (111) CD and $(11\bar{1})CD$ planes can connect to the (0001) lattice of compressed graphite. Therefore, {0001} graphite to {111} diamond is the core transformation orientation relationship in graphite transferring to diamond. For the transformation process from

amorphous carbon to diamond, Li et al.² prepared the composite consisting of ultrafine nanodiamond dispersed in disordered graphene and indicated amorphous carbon transforms into diamond through a nucleation process via a local rearrangement of carbon atoms and diffusion-driven growth. In this process, the diamond crystallites are evenly in situ generated by disordered carbon fragments. These diamond crystals remain nano-sized. The multiple sub-twins can be observed on the low-energy $\{111\}$ planes of those nanodiamonds. In addition, fullerenes are capable of undergoing similar transformations at high temperature and high pressure (HTHP) conditions^{3,4}. The diamond-to-graphite transformation can be regarded as the reverse process of the graphite to diamond transformation. From this we suggest that the diamond to graphite transformation process may also produce intermediate phases with different sp^2 carbon phases. In this process, the sp^2 amorphous carbon phase may have local energy minima than graphite fragments. This is the reason for the amorphous carbon phases that can be observed in HRTEM image of CPD. The corresponding section in the manuscript also gives a more critical discussion:

The transformation process of diamond to graphite has been controversial. Studies^{1-4,13} have shown that there is a specific phase relation between graphite and diamond during the transformation process. The higher-magnification image (Fig. 1b) of the selected cyan zone in Fig. 1a also indicates the possibility of this phase transition process. The $(\bar{1}\bar{1}\bar{1})$ diamond plane oriented to the $[\bar{1}\bar{1}\bar{2}]$ zone axis (Fig. 1b marked blue) was directly transformed into the rectangular-like $\{0002\}$ graphite plane oriented to the $\langle 1010 \rangle$ zone axis (Fig. 1b, marked red 1) and subsequently into the fingerprint-like $\{0002\}$ graphite plane (Fig. 1b, marked red 2). This transformation process can be represented as $d\{111\} \rightarrow g\{0002\}$. This transformation process is similar to that of graphite to diamond^{1,2}. In addition, intermediate amorphous carbon phase (Fig. 1b, marked yellow) may be formed during the phase transition process^{3,4}. The amorphous carbon also continued to transform into graphite fragments to reach a more stable state (Fig. 1b, marked red). The selected-area electron-diffraction (SAED) image revealed a broad amorphous carbon halo and face-centered cubic diamond crystalline spots oriented to the $[\bar{1}\bar{1}\bar{2}]$ zone axis (Fig. 1c). In contrast, the graphite phase was not prominent, because of its low concentration and the overlapping of its $\{0002\}$ planes with the amorphous halo.

(4) Temperature resolution and measurement limit are two key parameters for temperature measurement. In addition, for rapid cooling and heating applications, the time resolution of temperature measurement is also extremely important. For this aspect, it may be necessary to supply some data or discussion.

Response: Thank you for the constructive suggestions provided by the reviewer. You emphasized the significance of time resolution in temperature sensing, particularly in the context of abrupt temperature variations. We wholeheartedly concur with your viewpoint. Time resolution indeed plays a pivotal role in ensuring the accuracy and reliability of temperature measurements. In rapid temperature change experiments, the response time of temperature sensors directly influences data accuracy. In light of this, we have conducted the following supplementary experiments and discussions regarding the time resolution of CPD.

We initially conducted a test for the temperature response time of the CPD in an abrupt temperature change process. A sample holder with a CPD sample was connected to a DMM7510 digital multimeter using pure silver wires in a 2-probe configuration, as illustrated in Video1. The sample was first set to equilibrate at room temperature. Subsequently, it was directly immersed in liquid nitrogen. Upon achieving a stable resistance reading on the Digit Multimeter, the sample was retrieved from the liquid nitrogen and placed back at room temperature, where it was left to stabilize once again. The testing procedure is depicted in Video1, and the R-T curve during this process is depicted in Fig. R2a. When the sample reached a stable state at room temperature (22.9°C), noted as T_0 , the average resistance was 87.0372 Ω , denoted as R_0 , and the corresponding time was recorded as t_0 . Upon rapid immersion in liquid nitrogen and complete submersed, the displayed average resistance stabilized at 126.6393 Ω , designated as R_1 , corresponding to the temperature of liquid nitrogen (-195.8°C), denoted as T_1 . The time difference t_0-t_1 , represents the duration required for the step-change temperature measurement from room temperature to the temperature of liquid nitrogen, which is 3.48 s. The R-T curve in Fig. R2a reveals three distinct stages after immersing the sample into liquid nitrogen. Initially, from t_0 to $t_{0.26}$, the resistance exhibits a relatively slow, nearly linear increase, taking approximately 1.74 s. Following this, there is a sudden, rapid increase in resistance within a very short period of 0.3 s ($t_{0.26}-t_{0.9}$). Subsequently, the resistance takes an extended duration of approximately 1.44 s to reach the ultimate steady state, occurring at $t_{0.9}-t_1$.

In general, the response of a temperature sensor can be expressed as an energy balance

between the rate of internal energy change within the sensor and the rate of heat transfer between the sensor and its surrounding environment¹⁴. According to relevant researches^{15,16}, the response of a temperature sensor to such a step-change in temperature should theoretically be exponential. However, this clearly contradicts the experimental data. This phenomenon can be attributed to the sudden temperature decrease when the sample is directly immersed from the air into liquid nitrogen, causing a significant amount of water vapor on the sample and holder surfaces to condense during this rapid cooling process. On the other hand, the curve change during the $t_{0.26}$ - t_1 phase is more in line with the response curve when the sensor undergoes direct heat exchange with the environment. In this case, the time required is only 1.74 s.

Fig. R2. The response time of CPD during rapid temperature changes. a, Response time of CPD when cooling from room temperature to liquid nitrogen temperature. b, Calculation of the thermal time constant of CPD.

Furthermore, we conducted an additional analysis of the thermal time constant (TTC) of CPD. TTC is a characteristic constant for the response time of thermistor material. The heat balance equation for an isotropic solid without internal heat sources can be represented as¹⁶:

$$-\rho \cdot C_p \cdot V dT = \alpha \cdot A \cdot (T - T_1) dt \quad (1)$$

As the direct measurement data in this paper is resistance, and resistance corresponds directly to temperature, all parameters related to T in the above equation have been replaced with R :

$$-\rho \cdot C_p \cdot V dR = \alpha \cdot A \cdot (R - R_1) dt \quad (2)$$

In Eq. (2), A represents the sample's surface area, ρ is the sample's density, C_p and V are the specific heat capacity and volume of the sample, respectively. α is the heat

transfer coefficient between the sample and the surrounding medium, and it can be influenced by various factors, including the shape and size of the sample, as well as the method of how the sample is contacting with the surrounding medium, the form and flow rate of the medium, and more.

When the sample is moved from air to liquid nitrogen, the resistance changes from R_0 to R_1 . The relationship between t and R at any given moment during this process can be obtained by integrating equation (2), resulting in:

$$\frac{R - R_1}{R_1 - R_0} = -e^{-\frac{t}{\tau}} \quad (3)$$

Where τ is the thermal time constant,

$$\tau = \frac{\rho C_p V}{\alpha A} \quad (4)$$

Eq. (3) can be further written as:

$$R_1 - R = (R_1 - R_0)e^{-\frac{t}{\tau}} \quad (5)$$

Taking the natural logarithm of both sides of Eq. (5), we can further express it as:

$$\ln(R_1 - R) = \ln(R_1 - R_0) - \frac{t}{\tau} \quad (6)$$

Clearly, plotting $\ln(R_1 - R)$ against t results in a linear relationship, where the slope of the line corresponds to $-\frac{1}{\tau}$, then get the value of τ . As shown in Fig. R2b, the curve is not a single line; instead, it is composed of two linear segments, which is due to the reasons mentioned earlier. We conducted separate curve fitting for the two segments, yielding τ_1 and τ_2 . τ_1 is 5.84 s, and τ_2 is 0.09 s. τ_1 is situated within the region that is significantly affected by the intermediate medium (water vapor), leading to a substantial overestimation of its value. In contrast, $t_{0.26}$ can still be reasonably regarded as the starting point of the step-change temperature. Therefore, τ_2 holds a greater reference value. We have provided supplementary explanations to the original manuscript as:

In addition, Supplementary Fig. 6 shows the response time of the CPD. Put the CPD into liquid nitrogen from room temperature until stable, and the corresponding response time is 3.48s. The time taken to respond to 90 % of a step change in temperature ($T_{0.9}$) is only 2.04s.

(5) As the authors mentioned, the research on enhancing the oxidation resistance of diamond often centers on diamond graphitization-prevention strategies. Interestingly,

this work found that the incorporation of an sp^2 -hybridized carbon phase within the diamond structure increases the onset oxidation temperature. An explanation should be given.

Response: We appreciate the reviewer's constructive suggestion. We think that these anomalous characteristics may be related to the structure of CPD. Evidence supporting this hypothesis can be derived from TGA/DSC analysis (Fig. R3) and SEM observation (Fig. R4 and R5).

Fig. R3. Thermogravimetric and differential scanning calorimetry curves of original diamond CPD, measured in air at a heating rate of $5 \text{ K} \cdot \text{min}^{-1}$. a, and c, Thermogravimetric curves. b, and d, Differential scanning calorimetry curves.

In general, the presence of sp^2 carbon reduces the resistance of diamond to high temperature oxidation. This is because their own onset oxidation temperature is much lower. Moreover, their presence accelerates the transformation of diamond to graphite, which in turn destroys the intact diamond crystal, further reducing the high-temperature oxidation resistance of diamond¹⁷. However, the microstructural characteristics of CPD is sp^3 carbon dominated (sp^3 carbon contents $> 50 \%$) and sp^2 nano carbon phase uniformly dispersed. This microstructure avoids the existence of larger continuous sp^2 carbon bonds and protects sp^2 nano carbon phase from directly contact with oxygen, which can inhibit the sp^2 carbon clusters from increasing at high temperatures¹⁸.

Fig. R6. The thermal expansion coefficient of CPD.

Moreover, the thermal expansion coefficient of sp^2 -hybridized carbon shows a significantly higher value than that of diamond^{19,20}, which means that CPD may be more susceptible to thermal expansion. The experimentally measured thermal expansion coefficient of CPD is 8.12×10^{-6} (Fig. R6), which is higher than the literature value of 2.91×10^{-6} for diamond²¹, and is consistent with the above conclusion. As shown in (Fig. R3b), compared to original diamond, the DSC curve of CPD shows a continuous slight exothermic process (marked green) during the heating up process. The exothermic phenomenon originates from the oxidation of the exposed sp^2 carbon phase and the thermal expansion of the sp^2 carbon phase surrounded by the sp^3 carbon phase²²⁻²⁴. The thermal expansion further causes internal stresses to increase, which is important for delaying the oxidation onset point of diamond²⁵⁻²⁷. There was no significant decrease in the TG curve during this process (Fig. R3a), implying that the CPD remained structurally intact and the sp^3 hybridized phase wasn't oxidized.

Fig. R4. Oxidation of original diamond at different temperatures. a, 700 °C, 3 min. b, 900 °C, 3 min. c,

1000 °C, 3 min. d, 1000 °C, 10 min. e, 1000 °C, 30 min. f, 1100 °C, 3 min.

Fig. R5. Oxidation of CPD at different temperatures. a, 700 °C, 3min. b, 900 °C, 3 min. c, 1000 °C, 3 min. d, 1000 °C, 10 min. e, 1000 °C, 30 min. f, 1100 °C, 3 min.

The SEM images more visually shows the differences in the oxidation processes of CPD and diamond. From 700°C onwards, obvious etch lines due to oxidation appeared on the original diamond (Fig. R4a). With the increase in oxidation temperature and oxidation time, the etching lines rapidly developed into larger and deeper etching pits, and the surface of the original diamond showed obvious damage (Fig. R4b-f). As for CPD, at and below 900°C, the surface of the CPD sample remained dense with some slight bumps and no traces of etching by oxidation (Fig. R5a and R5b). After heating up to 1000 °C, fine and short etching lines began to appear on the CPD surface (Fig. R5c). With the increase in oxidation time as well as the increase in oxidation temperature, the width and depth of the etching lines gradually increased (Fig. R5c-f). In addition, the etch lines on the CPD surface are very uniform at different oxidation temperatures and oxidation times. This also indirectly confirms that the discrete distribution of the sp^2 hybridized carbon phase can effectively delay the rapid oxidation of the sp^3 carbon phase, so that the whole oxidation process proceeds uniformly and slowly.

Since the high temperature oxidation resistance performance is not the focus of this work, we have made a minor modification in the manuscript as follows:

The **composite phase** structure and the corresponding stress fields could be the primary factor responsible for the enhancement of the diamond's **high-temperature oxidation resistance**²⁵⁻²⁷.

This finding contradicts the commonly held belief³² that the presence of sp^2 -hybridized carbon bonds reduces thermal stability. We suppose that these anomalous characteristics may be related to the structure of CPD. Compared to traditional graphitized diamond, the microstructural characteristics of CPD is sp^3 carbon dominated ($> 50\%$ sp^3 carbon) with sp^2 nano carbon phase uniformly discrete distributed. This microstructure avoids the existence of larger continuous sp^2 carbon bonds and protects sp^2 nano carbon phase from directly contacting with oxygen, which can inhibit the sp^2 carbon clusters from increasing at high temperatures¹⁸.

Moreover, the sp^2 carbon phase has a greater thermal expansion coefficient than that of diamond²¹. The thermal expansion of those sp^2 nano carbon phases makes the internal stresses of CPD further increase, which is the main factor to enhance the diamond's resistance to high-temperature oxidation resistance performance. The SEM images show the differences in the oxidation processes of diamond and CPD ((Fig. R4 and Fig. R5) more visually.

(6) What is the sample size used in Figs. 2 to 4? In other words, does the sample size affect the performance of the temperature-sensing characteristics? The details of sample synthesis, such as the sintering method, equipment model, and sintering pressure, should be given. The authors also should provide detailed descriptions of the 3D-printed current structures in the Method section.

Response: We appreciate the reviewer's constructive suggestion. The sample size corresponding to Fig. 2-4 is $5 \times 5 \text{ mm}^2$. As you mentioned, different sample sizes may result in variations in performance. In this work, variations in sample sizes have led to differences in resistance. The larger the sample size, the higher its resistance. However, the resistance itself does not affect the temperature-sensitive performance of CPD. Variations in sample size do not impact the characteristics of CPD's NTC or the changes in its R-T curve pattern. We present the R-T curves for different batches of CPD samples in Fig. R7. All the samples described below were fabricated using 3D printing, and the specific fabrication process is detailed in the "Methods" section. The sample sizes vary as follows: sample A ($8 \times 4 \text{ mm}^2$), sample B ($4 \times 4 \text{ mm}^2$), sample C ($8 \times 5 \text{ mm}^2$), sample D ($10 \times 5 \text{ mm}^2$), and sample E ($5 \times 5 \text{ mm}^2$). While sample B has the smallest surface area, its resistance at room temperature and cryogenic temperatures is slightly higher than that of sample E. Additionally, sample A exhibits lower resistance at room temperature

compared to sample C and sample D. However, as the temperature decreases, the rate of resistance increases in sample A is significantly higher than that of sample C and sample D, resulting in the highest resistance among all samples at 2K. Currently, there is no definitive explanation for the observed differences, and it's worth noting that this phenomenon is not unique to CPD. Taking the widely recognized *Cernox*TM cryogenic temperature sensor, for example, each of its samples exhibits distinct R-T curves and requires individual calibration. In the low-temperature range, the resistance differences of *Cernox*TM cryogenic temperature sensors can span two orders of magnitude. In addition, the other characteristics of these CPD samples, such as negative temperature coefficient, high resistivity, and the trends in R-T curve variations, all align with the conclusions presented in the manuscript. Therefore, we consider such variations of CPD samples to be entirely reasonable and acceptable, and they, to some extent, corroborate the high repeatability of CPD. Furthermore, all R-T curves, when fitted with the ExpDec3 function, yield R² values exceeding 0.9999. For practical temperature sensor production, this will significantly reduce the workload involved in sensor calibration. The primary reason for not further verifying larger-sized samples is due to constraints imposed by the sample holder dimensions. Additionally, smaller-sized samples tend to exhibit greater sensitivity to temperature changes and are more space-efficient in practical usage scenarios.

Fig. R7. R-T curves and of CPD samples with different sizes: sample A (8×4 mm²), sample B (4×4 mm²), sample C (8×5 mm²), sample D (10×5 mm²), and sample E (5×5 mm²).

We have provided additional details regarding the experimental procedures, as follows:

The diamond particles (80μm) were fully mixed with acrylic acid ammonium salt

polymer ($\geq 99\%$), acrylamide ($\geq 99\%$), N,N'-Methylenebisacrylamide ($\geq 99\%$), 2-hydroxy-2-methylpropiophenone ($\geq 99\%$), diphenyl(2,4,6-trimethylbenzoyl)phosphine oxide ($\geq 99\%$) and water in a specific weight ratio of 60:3.3:2.4:4:0.24:7.3. 3D printing was carried out using a DIW 3D printer (3D Bio-Architect® Work Station) from Regenovo Biotechnology Co., Ltd., China. The printing was carried out with a line width of 0.4 mm, utilizing a closely spaced pattern that can be achieved through a crosshatch printing method. During the printing process, preliminary curing was accomplished by exposure to a 365 nm ultraviolet lamp. Then the sample was dried at 80 °C in an oven (Froilabo, France) for 4 hours. Subsequently, it was sintered in an Ar atmosphere at 1250 °C and atmospheric pressure to obtain the CPD. All samples were ultrasonically cleaned in anhydrous ethanol. The sintering process is conducted in a tube furnace (BTF-1700C, ANHUI BEQ EQUIPMENT TECHNOLOGY CO., LTD., China). Unless otherwise specified, the sintering time for the samples presented in the manuscript is 1800 min.

Minor:

(1) In the introduction, the authors said that “(Line 48) However, compared with diamond, sp²-hybridized carbon materials are fragile and unstable.” This statement is inappropriate. Graphite is a typical sp²-hybridized carbon material, and it is more stable than diamond at room temperature and standard atmospheric pressure.

Response: We thank the reviewer for the suggestion. Indeed, as you mentioned, graphite is a carbon allotrope with greater chemical stability at room temperature. The confusion may stem from our previous expression. Here, we intend to convey that due to graphite's layered structure and the weaker van der Waals forces between the layers, graphite exhibits a softer texture and significantly inferior mechanical properties compared to diamond. It is prone to flaking or getting damaged. We have made the necessary revisions in the manuscript, which now read as follows:

In comparison to diamond, graphite has a softer texture and inferior mechanical properties, making it highly susceptible to flaking or getting damaged during use.

(2) In this work, the amorphous carbon has an average size of 3.55 nm and the graphite fragment has an average size of 3.01 nm. What is the thickness of the TEM sample in Fig. 1? Generally, the thickness of the TEM sample is about tens to hundreds of nanometers. If so, a misjudgment of their sizes could result from the overlap of the

amorphous carbon and graphite.

Response: Thank you for the feedback. The thickness of the TEM samples in the manuscript is approximately 20-30 nm. As the reviewer correctly pointed out, the thickness of the samples can lead to inaccuracies in determining the size of the amorphous carbon and graphite fragments, especially in regions where amorphous carbon and graphite fragments are adjacent. Therefore, the measured size for amorphous carbon zone and graphite fragments zone should be only considered as a rough estimate of the phase distribution within the CPD sample. In light of this, we have clarified the original manuscript accordingly:

The graphite fragments zone with a size of 3.01 nm (17% volume fraction) showed an interatomic spacing of 0.338 nm, which corresponded to the graphite fragments {0002} at the arced position of the FFT image with a brighter contrast (Fig. 1a inset). It should be noted that the thickness of the TEM sample may result in inaccuracies in the analysis of the delineation of the amorphous carbon and graphite fragments. Therefore, these values are only provided as a rough reference for the distribution trends of the amorphous carbon zone and graphite fragments zone.

(3) The calculation details of the phase content should be given.

Response: Thank you for the reviewer's feedback. To depict the distribution of the amorphous carbon zone and graphite fragments zone in CPD, we utilized multiple HRTEM images, including those in the manuscript, for false-color calibration of different phase regions. We then employed ImageJ software to quantitatively analyze the corresponding areas, ultimately determining the content of each phase.

(4) The authors' previous works and numerous previous studies (such as Matter 2023, 6, 7) have shown us that designing a dual-phase structure is a powerful approach to improving the mechanical properties of materials. However, it is difficult to imagine from these works that a dual-phase material can overcome the bottleneck of low-temperature measurement. Therefore, a related description should be supplied in the introduction.

Response: We thank the reviewer for pointing out the oversight in the introduction. We have provided supplementary explanations to the original manuscript as:

Furthermore, the combination of sp^2 hybridized phases with diamond can significantly impacts the electrical conductivity, bandgap, and other characteristics of diamond-based materials^{28,29}. This approach offers insights into the potential applications of diamond in the field of electronics.

(5) I can't see the specific text in the inset table in Fig. 2.

Response: Thank you for the reviewer's feedback. We will place the table separately here and enlarge it within the main text.

Table R2. The fitting equation for the R-T curve in Fig. 2a.

Equation	$y = A_1 \cdot \exp(-x/t_1) + A_2 \cdot \exp(-x/t_2) + A_3 \cdot \exp(-x/t_3) + y_0$
y_0	3.54257 ± 0.13585
A_1	0.71116 ± 0.00924
t_1	1.95884 ± 0.01426
A_2	11.52731 ± 0.13094
t_2	1616.04264 ± 24.19593
A_3	0.58585 ± 0.00479
t_3	87.731 ± 0.52091
R^2	0.99999
Adj. R^2	0.99999

(6) Although the details of the strength test are described in the method, the relevant data are not given in the manuscript.

Response: Thank you for the reviewer's feedback. We characterized the mechanical properties of the samples, but we believe that the experiments conducted in this regard were not comprehensive enough. To maintain the rigor of the paper, we have removed this section from the main part of the manuscript. However, we inadvertently overlooked the description in the methods section, which has also been removed from the manuscript.

References

1. Luo, K. et al. Coherent interfaces govern direct transformation from graphite to diamond. *Nature* **607**, 486-491 (2022).
2. Li, Z. et al. Ultrastrong conductive in situ composite composed of nanodiamond incoherently embedded in disordered multilayer graphene. *Nat. Mater.* **22**, 42-49 (2023).
3. Zhang, S. et al. Discovery of carbon-based strongest and hardest amorphous material. *Natl. Sci. Rev.* **9**, nwab140 (2022).
4. Zhang, S. et al. Narrow-gap, semiconducting, superhard amorphous carbon with

- high toughness, derived from C60 fullerene. *Cell Reports Phys. Sci.* **2**, 100575 (2021).
5. Hui, W. et al. Red-Carbon-Quantum-Dot-Doped SnO₂ Composite with Enhanced Electron Mobility for Efficient and Stable Perovskite Solar Cells. *Adv. Mater.* **32**, 1-9 (2020).
 6. Jubu, P. R. et al. Dispensability of the conventional Tauc's plot for accurate bandgap determination from UV-vis optical diffuse reflectance data. *Results Opt.* **9**, 0-6 (2022).
 7. Crnjac, A. et al. Energy loss of MeV protons in diamond: Stopping power and mean ionization energy. *Diam. Relat. Mater.* **132**, 109621 (2023).
 8. Oku, T. Carbon/Carbon Composites and Their Properties. in Carbon Alloys (eds. Yasuda, E. et al.), Ch. 33, (Elsevier Science, 2003).
 9. Park, S. J., & Seo, M. K. Types of Composites. *Interface Sci. Technol.*, **18**, 501-629 (2011).
 10. Robertson, J. Diamond-like amorphous carbon. *Mater. Sci. Eng. R Reports* **37**, 129-281 (2002).
 11. Lin, Y., Zhao, Z., Strobel, T. A. & Cohen, R. E. Interpenetrating graphene networks: Three-dimensional node-line semimetals with massive negative linear compressibilities. *Phys. Rev. B* **94**, 1-9 (2016).
 12. Jiang, X., Zhao, J., Li, Y. L. & Ahuja, R. Tunable assembly of sp³ cross-linked 3D graphene monoliths: A first-principles prediction. *Adv. Funct. Mater.* **23**, 5846-5853 (2013).
 13. Németh, P. et al. Diamond-graphene composite nanostructures. *Nano Lett.* **20**, 3611-3619 (2020).
 14. Rinaudo, P., Paya-Zaforteza, I., Calderón, P. & Sales, S. Experimental and

- analytical evaluation of the response time of high temperature fiber optic sensors. *Sens. Actuator A Phys.* **243**, 167-174 (2016).
15. Dzinavatonga, K., Obileke, K. C., Makaka, G., Mukumba, P. & Munyaradzi Nyambo, B. Determination of Thermal Time Constant of a Pt100 Temperature Sensor in Still Oil Using the Time Derivative Method. *Chem. Eng. Technol.* **46**, 1673-1678 (2023).
 16. Wen, J. H. et al. Response Time of Microfiber Temperature Sensor in Liquid Environment. *IEEE Sens. J.* **20**, 6400-6407 (2020).
 17. Joshi, A., Nimmagadda, R. & Herrington, J. Oxidation kinetics of diamond, graphite, and chemical vapor deposited diamond films by thermal gravimetry. *J. Vac. Sci. Technol. A: Vac. Surf. Film.* **8**, 2137-2142 (1990).
 18. Kalish, R., Lifshitz, Y., Nugent, K. & Praver, S. Thermal stability and relaxation in diamond-like-carbon. A Raman study of films with different sp³ fractions (ta-C to a-C). *Appl. Phys. Lett.* **74**, 2936-2938 (1999).
 19. Champi, A., Lacerda, R. G., Viana, G. A. & Marques, F. C. Thermal expansion dependence on the sp² concentration of amorphous carbon and carbon nitride. *J. Non. Cryst. Solids* **338-340**, 499-502 (2004).
 20. Zhao, L., Tang, J., Zhou, M. & Shen, K. A review of the coefficient of thermal expansion and thermal conductivity of graphite. *New Carbon Mater.* **37**, 544-555 (2022).
 21. Krishnan, R. Thermal Expansion of Diamond. *Nature* **154**, 486-487 (1944).
 22. Hakovirta, M., Vuorinen, J. E., He, X. M., Nastasi, M. & Schwarz, R. B. Heat capacity of hydrogenated diamond-like carbon films. *Appl. Phys. Lett.* **77**, 2340-2342 (2000).
 23. Haddon, R. C. & Chow, S. Y. Hybridization as a metric for the reaction

- coordinate of the chemical reaction. Concert in chemical reactions. *Pure Appl. Chem.* **71**, 289-294 (1999).
24. Kane, J. J., Contescu, C. I., Smith, R. E., Strydom, G. & Windes, W. E. Understanding the reaction of nuclear graphite with molecular oxygen: Kinetics, transport, and structural evolution. *J. Nucl. Mater.* **493**, 343-367 (2017).
 25. Yan, X. et al. High temperature surface graphitization of CVD diamond films and analysis of the kinetics mechanism. *Diam. Relat. Mater.* **120**, 108647 (2021).
 26. Enriquez, J. I. et al. Oxidative etching mechanism of the diamond (100) surface. *Carbon.* **174**, 36-51 (2021).
 27. Huang, Q. *et al.* Nanotwinned diamond with unprecedented hardness and stability. *Nature* **510**, 250-253 (2014).
 28. Yang, N. et al. Conductive diamond: Synthesis, properties, and electrochemical applications. *Chem. Soc. Rev.* **48**, 157-204 (2019).
 29. Li, H. et al. Tailoring the sp²/sp³ carbon composition for surface enhancement in Raman scattering. *Appl. Surf. Sci.* **599**, (2022).

Reviewer #2:

The authors successfully synthesized a DPD with remarkable electrical properties and thermal stability through a straightforward approach. The DPD seems to demonstrate its capability and stability as a temperature sensing material in extreme environments, implying the potential application to the low-temperature physics and quantum physics. In my opinion, most of the contents of the manuscript are well organized and easy to read in different fields. In particular, this work may extend the application of diamond from a superhard material to a functional material. However, some critical issues prevent the paper from being published in its current version.

Response: We appreciate the positive evaluation provided by the reviewer and are thankful for the thorough and insightful comments. Your feedback has greatly contributed to improving the quality of our work. We have carefully addressed all the comments from the reviewer and provided detailed responses as outlined below:

1. The authors made a strong claim that the lowest temperature measurement limit (in fact, it is a prediction) of the DPD was 1 mK based on the true low temperature measurement down to 2K. Meanwhile, in the introduction, the authors mentioned that “traditional thermometers can only measure temperatures higher than 20 mK”. So I asked the authors to do additional experiments at the current limit (e.g., about 20 mK or slightly higher) and repeated the measurements by using several DPD samples (to exclude the effect of size and quality of samples). In the published paper [Chem. Mater. 33, 8722 (2021); <https://doi.org/10.1021/acs.chemmater.1c02683>], the temperature-dependent resistivities of Ti_2Co shows the semiconducting behavior from the temperature range of 300 K to 2 K [see Fig 4c and 4d]. Strikingly, it shows superconductivity at ~ 0.75 K. In this case, some materials exhibit abnormal transition at a very low temperature beyond what we usually expected. From this point, it is a premature conclusion that the lowest temperature measurement limit of DPD was 1 mK without the solid experimental evidence (only by using an extrapolation method). If the authors could not provide further/new experimental evidence in the revision, I suggest the author to revise their conclusion to “the predicted temperature measurement limit of the DPD reaches the mK-level” and transfer to another journal.

Response: We thank the reviewer for the suggestion. According to comments from another reviewer, the term ‘Dual-Phase Diamond’ (DPD) has been replaced with

‘Diamond with sp^2 - sp^3 composite phase’ (CPD) in the manuscript. Therefore, all ‘DPD’ have been replaced with ‘CPD’ in subsequent responses. Indeed, as stated by the reviewer, only obtaining test data at the temperature of 2 K is not sufficient to prove that CPD can be used at mK-level temperatures. Therefore, we have characterized the resistance properties of CPD at mK-level temperatures with dilution refrigerator in additional experiments, and data from multiple samples are listed to demonstrate the reproducibility of CPD performance. The corresponding experimental parameters have also been updated in the manuscript. Fig. R1a shows the CPD's resistance variations with temperature changes from 40 mK to 1 K and a comparison with the resistance value at room temperature. It can be seen that the resistance of the CPD also remains monotonically increasing over the 40mK-1K temperature interval, with no inflection points or abrupt changes. This means that the physical properties of the CPD remain stable when it approaches absolute 0, with its resistance varying within the expected range. The curve obtained after fitting the data using ExpDec3 is shown as the red curve in Fig. R1a. The curve is a good fit to the data with an R^2 of 0.9978, demonstrating that the samples also conform to the conclusions in the manuscript at temperatures of mK-level. It should be noted that due to the inherent characteristics of the dilution refrigerator, the cooling process is rapidly variable and unstable in the temperature range from room temperature to the K-level¹. Therefore, it is usually impossible to obtain accurate resistance values for samples in this temperature range by direct measurement. As a result, the fitted curve obtained in Fig. R1a will have a larger error than that obtained from the PPMS test. It's only used to demonstrate the reliability of the CPD's NTC performance at lower temperatures. Fig. R1b further shows details of the CPD resistance change between 40 mK-800 mK. At temperatures as low as 40mK, the resistance of the CPD rises to 11.27749 Ω . The CPD's resistance maintains a small monotonic increase as the temperature decreases and has a very low heat capacity (Fig. R3). The above characteristics allow the CPD to have a sensitive response to temperature changes without generating a large amount of self-heating, which is very favorable for temperature measurement at very low temperatures. We have also provided supplementary explanations to the original manuscript as follows:

Furthermore, a dilution refrigerator was used to characterize the change in resistance of CPD below 1K. As shown in Fig. R1a and R1b, the resistance of the CPD also remains monotonically increasing over the 40mK-1K temperature interval, with no inflection points or abrupt changes. In addition, the R-T curve of the CPD below 1 K has a

similarly high R^2 of 0.9978 when fitted using the Expdec3 function (Fig. R1a). The above results show that the physical properties of the CPD remain stable when it approaches absolute 0 from room temperature.

Fig. R1. Resistance changes of CPD below 1 K measured by a dilution refrigerator. a, R-T curve from 40 to 1000 mK of the CPD sample; black dots: experimental data; red line: data fitted using the Expdec3 function. b. Enlarged view of the data from 40 mK - 800 mK.

Further, to verify the reproducibility of the CPD samples, we have additionally provided R-T data for 5 samples of different sizes (Fig. R2). In fact, as suggested by the reviewer, we had planned to conduct this experiment in the PPMS as well as the dilution refrigerator. However, due to the fact that all samples had to be prepared separately, the huge amount of work, the long experimental period and our unskilled experimental skills in the dilution refrigerator prevented us from doing so. Therefore, only the resistances of the different samples tested in the PPMS are presented here. Even so, we believe that these data are sufficient to demonstrate the reliability and high reproducibility of the CPD. Among them, sample A has dimensions of $8 \times 4 \text{ mm}^2$, sample B has dimensions of $4 \times 4 \text{ mm}^2$, sample C has dimensions of $8 \times 5 \text{ mm}^2$, sample D has dimensions of $10 \times 5 \text{ mm}^2$, and sample E has dimensions of $5 \times 5 \text{ mm}^2$. It can be observed that there are slight differences in resistance values among samples of different sizes. On one hand, the differences in resistance are expected due to the varying sizes of the samples. On the other hand, manual operations inherently introduce various unavoidable sources of error, so these variations are understandable. Furthermore, such differences are common in NTC materials. Taking the renowned Cernox temperature sensor as an example, the resistance values of different samples at the same temperature can vary by up to 2 orders of magnitude. Furthermore, their other characteristics,

including the negative temperature coefficient, low-temperature behavior, cyclic stability, and fitting characteristics, all align with the conclusions presented in the manuscript.

Fig. R2. R-T curves and of CPD samples with different sizes: sample A ($8 \times 4 \text{ mm}^2$), sample B ($4 \times 4 \text{ mm}^2$), sample C ($8 \times 5 \text{ mm}^2$), sample D ($10 \times 5 \text{ mm}^2$), and sample E ($5 \times 5 \text{ mm}^2$).

In summary, combined with the data available to date, we believe that the above experimental results are sufficient to demonstrate that the CPD is capable of temperature measurement at temperatures on the mK-level, and that this property is highly repeatable.

Minor questions:

2. How did the authors determine the exact temperature value and collect the data in a range of 2 mK as shown in Extended Data Fig. 4d? Furthermore, what standard was used to divide the data into two intervals?

Response: Thank you for the reviewer's feedback. The display accuracy of PPMS[®] DynaCool[™] of Quantum Design Inc., after calibration, can achieve up to six decimal places. In the dynamic response test, we programmed the system to cyclically increase and decrease the chamber temperature at a fixed rate of $0.5 \text{ K} \cdot \text{min}^{-1}$ around the 3 K temperature point. We recorded the relevant readings. While we may not precisely control the chamber temperature to exactly 3.001 K or 2.999 K, through the continuous cyclic temperature changes, we were able to capture the continuous temperature variation curve and the corresponding CPD resistance values. Subsequently, we

performed statistical analysis on the resistance values falling within this temperature intervals. We collected a total of 852 data points and further divided these data points into two intervals based on the displayed readings from the PPMS: $2.999 \leq T < 3 \text{ K}$ and $3 \leq T < 3.001 \text{ K}$.

Analyzing the resistance values within these intervals allowed us to assess the data's degree of dispersion and ultimately confirm that there is a significant difference in resistance values between the two temperature intervals. After completing the mentioned filtering, we conducted various statistical analyses on the resistance data within the two intervals, including calculating the mean, variance, and standard deviation. As shown in Table R1, the variance and standard deviation of these two datasets are significantly smaller than the mean. The standard deviation is 5 orders of magnitude smaller than the mean, and the variance is 9 orders of magnitude smaller. From a statistical perspective, this is a sufficient evidence to establish that both sets of data exhibit an extremely high level of concentration. Furthermore, we also calculated the 95% confidence intervals for these two sets of data separately. As shown in Table 1, these two intervals' 95% confidence intervals do not overlap. Furthermore, after conducting a one-way analysis of variance (ANOVA) between both sets of data, shown in Table R1 we obtained a p-value of 6.9×10^{-6} , which is significantly smaller than 0.001. In terms of statistical significance, this represents an extremely significant result. This indicates that when the temperature is within the two temperature intervals mentioned above, there are distinct differences in the CPD resistance values.

Table R1 One-way analysis of variance of selected data sets.

source of variation	SS	df	MS	F	P-value	F crit
Between-group variation	1.13×10^{-6}	1	1.13×10^{-6}	20.47595	6.90×10^{-6}	3.852422

3. From the data shown in the paper, the resistance values are consistent at the same temperature. Are these data obtained from the only one sample? Would the measured values change with the quality and size of the DPD samples?

Response: We thank the reviewer for the constructive suggestion. As you mentioned, the size of the samples can indeed influence the experimental results. However, the difference in resistance due to size does not affect the NTC characteristics of the CPD. This characteristic has been demonstrated in the previous section (Fig. R2) and will not

be repeated here.

4. Could the authors theoretically explain how the DPD works as a cryogenic temperature sensor to some extent beside using a three-phase exponential decay function.

Response: Thank you for the reviewer's feedback. We'd like to provide a detailed explanation here of why CPD is suitable for use as a temperature sensor and make corresponding revisions to the manuscript.

The fundamental principle of a temperature sensor is that a specific property of a material changes with temperature. By recording this property, the ambient temperature of the sensor's surroundings can be determined. In theory, within a specific temperature range, any material with a monotonically changing property with temperature can be used as a temperature sensor in that temperature range, such as thermal expansion, resistance change, thermoelectric effects, and more². Resistance change is the most widely utilized characteristic for temperature sensing, and NTC (Negative Temperature Coefficient) materials are particularly suitable for extremely low-temperature measurements. This is because as the temperature decreases, the resistance of NTC materials increases, allowing them to sensitively indicate temperature variations³. NTC materials generally have the characteristic of relatively high resistance at cryogenic temperatures. The advantage lies in their ability to accurately respond to temperature changes, while the disadvantage is that materials with high thermal capacity and resistance tend to generate self-heating at cryogenic temperatures, which can affect the accuracy of low-temperature measurements. Additionally, materials with high resistance require larger operating power. In general, when the resistance exceeds 10,000 Ω , it is no longer suitable to be used as a cryogenic temperature sensor. As noted earlier in the manuscript and in this response, CPD is a kind of diamond-based material. The resistance of the CPD is only a few tens of ohms and increases monotonically by a small percentage as the temperature decreases. Compared to most cryogenic sensors with resistance values of thousands of ohms, the CPD has a clear advantage. In addition, we further characterized the coefficient of thermal expansion and heat capacity of the CPD as well, as shown in Fig. R3. At cryogenic temperatures, the CPD has an extremely low coefficient of thermal expansion and low specific heat capacity, which is close to that of diamond⁴. Therefore, CPD is an NTC material with low resistance and low specific heat. These properties above can make CPD more sensitive to temperature changes at cryogenic temperatures, as well as lower self-heating and less power loss.

Fig. R3. Thermal properties of CPD. a, The thermal expansion coefficient of CPD. b, The heat capacity of CPD.

In addition, compared with the expensive cost of precious metal-based materials, such as RuO₂, the raw material of CPD is synthetic diamond. The cost disparity between these two materials ranges from tenfold to several hundredfold. The significant economic value in practical production cannot be ignored. In summary, CPD is an ideal material for cryogenic temperature measurement.

References

1. Zu, H., Dai, W. & Waele, A. T. A. M. De. Development of dilution refrigerators - A review. *Cryogenics*, **121**, 103390 (2022).
2. Yeager, C. J. & Courts, S. S. A review of cryogenic thermometry and common temperature sensors. *IEEE Sens. J.* **1**, 352-360 (2001).
3. Chen, C. Evaluation of resistance-temperature calibration equations for NTC thermistors. *Meas. J. Int. Meas. Confed.* **42**, 1103-1111 (2009).
4. Vasil'ev, O. O., Muratov, V. B. & Duda, T. I. The study of low-temperature heat capacity of diamond: Calculation and experiment. *J. Superhard Mater.* **32**, 375-382 (2010).

Reviewer #3:

This paper introduces a dual-phase diamond (DPD) as a potential thermometer to probe temperatures below millikelvin levels. The authors synthesized DPD through the sintering of diamond powders with specific chemicals. Utilizing various analytical techniques, they confirmed the presence of graphitic phases (along with amorphous transition phases) within the diamond structures, resulting in the formation of DPD. The temperature dependence of the resistance of the produced DPD was measured, revealing a negative temperature coefficient trend. Extrapolating this dependence led them to conclude that the theoretical detection temperature of DPD could reach 1 mK, a level hitherto unattained by existing cryogenic temperature sensors. While I find the observed temperature range and magnetic susceptibility of DPD intriguing, I must emphasize that the experimental methods and data analysis demand a more rigorous treatment for this work to merit the aforementioned statement, as described below.

Response: Thank you for the objective evaluation of our work and the rigorous requirements for the experimental data. This forms the foundation for any research endeavor. We will provide a detailed response to the reviewers' comments and present the experimental data more comprehensively and in greater detail.

Firstly, the current dataset does not validate the operation and usefulness of thermometry at millikelvin temperatures. The data fitting was restricted to the range of 3-400 K, and all discussions related to the millikelvin thermometer operation rely on extrapolating the fitting curve to temperatures below 3 K. For instance, a study involving RuO₂-based thermometers for millikelvin temperatures (Cryogenics (Guildf). 32, 1167 (1992)), also cited as Ref. 32, experimentally demonstrated the R-T dependence down to 50 mK, which is two orders of magnitude lower than the lower limit experimentally confirmed in the present manuscript. Significantly, the data in Ref. 32 displays a plateau in the R-T curve below 0.1 K. This deviation from the conventional exponential decay dependence strongly implies the necessity of experimental verification of thermometry within the claimed temperature range. The exponential decay tendency of the present DPD cannot be assumed to behave straightforwardly.

Response: We thank the reviewer for the constructive suggestion. According to comments from another reviewer, the term 'Dual-Phase Diamond' (DPD) has been

replaced with ‘Diamond with sp^2 - sp^3 composite phase’ (CPD) in the manuscript. Therefore, all ‘DPD’ have been replaced with ‘CPD’ in subsequent responses. Indeed, as the reviewer notes, the difference between the expected use temperature in the manuscript and the actual temperature tested is three orders of magnitude. This is not enough evidence to show that CPDs can be used at temperatures on the mK-level. Therefore, we have characterized the resistance properties of CPD at mK-level temperatures with dilution refrigerator. Fig. R1a shows the CPD's resistance variations with temperature changes from 40 mK to 1 K and a comparison with the resistance value at room temperature. It can be seen that the resistance of the CPD also remains monotonically increasing over the 40 mK-1 K temperature interval, with no inflection points or abrupt changes. This means that the physical properties of the CPD remain stable as it approaches absolute 0, with its resistance varying within the expected range. The curve obtained after fitting the data also using ExpDec3 is shown as the red curve in Fig. R1a. The curve is a good fit to the data with an R^2 of 0.9978, demonstrating that the samples also conform to the conclusions in the manuscript at temperatures of mK-level. Fig. R1b further shows details of the CPD resistance change between 40 mK-800 mK. At temperatures as low as 40mK, the resistance of the CPD rises to 11.27749 Ω .

Fig. R1. Resistance changes of CPD below 1 K measured by a dilution refrigerator. a, R-T curve from 40 to 1000 mK of the CPD sample; black dots: experimental data; red line: data fitted using the Expdec3 function. b. Enlarged view of the data from 40 mK - 800 mK.

The CPD's resistance maintains a small monotonic increase as the temperature decreases and has a very low heat capacity (Fig. R3b). Even in temperature intervals below 100mK, the CPD's resistance change remains continuous and uniform, with no plateau temperatures or sudden changes. The above characteristics allow the CPD to

have a sensitive response to temperature changes without generating a large amount of self-heating, which is very favorable for temperature measurement at very low temperatures. We have also provided supplementary explanations to the original manuscript as follows:

Furthermore, a dilution refrigerator was used to characterize the change in resistance of CPD below 1 K. As shown in Fig. R1a and R1b, the resistance of the CPD also remains monotonically increasing over the 40mK-1K temperature interval, with no inflection points or abrupt changes. In addition, the R-T curve of the CPD below 1 K has a similarly high R^2 of 0.9978 when fitted using the Expdec3 function (Fig. R1a). The above results show that the physical properties of the CPD remain stable when it approaches absolute 0 from room temperature.

Secondly, the details regarding sample preparation and measurement methods are rather unclear. The methodology for sample preparation lacks several key points. For example, the type of diamonds used (Type-1b, 2a), information about impurity concentrations, whether HPHT or CVD diamonds were used, etc., remain unspecified. Furthermore, the phrase "mixed in a specific ratio" with chemicals (Line 274-281) requires clarification. In the measurement section, critical information such as the size of the measured sample (which could affect resistance measurements) and the methodology employed for making electrical contacts in the R-T data are absent. Is a four-point probe used even for low-temperature measurements? Additionally, it is mentioned that they fabricated micron-sized structures using FIB; however, it is important to note that the mere fabrication of micron-sized structures does not necessarily imply that these structures of DPD can effectively function as thermometers. These technical aspects are crucial for enabling the reproducibility of data by other researchers in the future. It is also noteworthy that the present paper only reports data from a single sample for R-T analysis. Thus, questions arise about reproducibility and potential variations in device fabrication.

Response: Thank you for the reviewer's thorough suggestions. We have provided a more detailed description of the sample preparation process and parameters, and made corresponding modifications to the original manuscript. The specific responses are as follows:

The original material of the CPD mentioned in the paper is HTHP diamond powder with an average particle size of 80 μm , which is obtained after 3D printing and sintering.

The specific preparation details are as follows:

The diamond particles (80 μ m) were fully mixed with acrylic acid ammonium salt polymer ($\geq 99\%$), acrylamide ($\geq 99\%$), N,N'-Methylenebisacrylamide ($\geq 99\%$), 2-hydroxy-2-methylpropiophenone ($\geq 99\%$), diphenyl(2,4,6-trimethylbenzoyl)phosphine oxide ($\geq 99\%$) and water in a specific weight ratio of 60:3.3:2.4:4:0.24:7.3. 3D printing was carried out using a DIW 3D printer (3D Bio-Architect® Work Station) from Regenovo Biotechnology Co., Ltd., China. The printing was carried out with a line width of 0.4 mm, utilizing a closely spaced pattern that can be achieved through a crosshatch printing method. During the printing process, preliminary curing was accomplished by exposure to a 365 nm ultraviolet lamp. Then the sample was dried at 80 °C in an oven (Froilabo, France) for 4 hours. Subsequently, it was sintered in an Ar atmosphere at 1250 °C and atmospheric pressure to obtain the CPD. All samples were ultrasonically cleaned in anhydrous ethanol. The sintering process is conducted in a tube furnace (BTF-1700C, ANHUI BEQ EQUIPMENT TECHNOLOGY CO., LTD., China). Unless otherwise specified, the sintering time for the samples presented in the manuscript is 1800 min.

It should be noted that 3D printing was used for shaping convenience and was not a necessary step in the production of CPD. The diamond powder is a commercially available synthetic diamond powder provided by Tianjian Carbon Material Co., Ltd., China, and belongs to the Type 1b diamond. The impurity content in the powder was calibrated using an oxygen-nitrogen-hydrogen analyzer (ONH836, LECO Corporation, USA) and an Inductively Coupled Plasma Optical Emission Spectrometer (ICP-OES, ULTIMA2, HORIBA Ltd., Japan). The results are shown in Table R1, and they are in good agreement with the data provided by the supplier.

Table R1 Impurity composition of original diamond.

Concentrations of impurity (ppm)	N	O	B	Al	Fe
Supplier provided	146	110	<1	2	18
Measured	200	60	/	6	20

The room temperature electrical conductivity of the CPD was measured using a four-point probe meter (CXT2665, Changzhou Xinyang Electronic Technology Co., Ltd., China), with a distance of 1.59 mm between the probes. The sample size used for testing

was no smaller than $10 \times 10 \text{ mm}^2$. The sample surface needs to be flat, ensuring that the four probes are in the same plane as the sample. The resistance values of the CPD at cryogenic temperatures and in a magnetic field were measured using a physical property measurement system (Quantum Design PPMS® DynaCool™, Quantum Design Inc., USA) equipped with a Cernox™ temperature sensor. Copper or silver wires were used to connect the sample holder and the sample, and the leads were secured with silver conductive paste (Solids Content $\geq 68 \text{ wt.}\%$).

Furthermore, we fully agree with the reviewer's comment that the mere fabrication of micron-sized structures does not necessarily imply that these structures of CPD can effectively function as thermometers. At this stage, we have only demonstrated that CPD can be processed and used at multiple scales. Creating micro-temperature sensors from CPD through FIB processing is a separate, complex endeavor, and our current work cannot substantiate this feature. Therefore, we have made revisions in the original manuscript to reflect this clarification:

These samples of different sizes can be applied to different occasions, enhancing the potential application value of CPD.

Additionally, the reviewer expressed concerns about the single-sample data. We fully agree with the reviewer's comments. We conducted multiple validations to ensure data reproducibility before drawing the conclusions in the original manuscript. In the original manuscript, we presented data from a single representative sample as we wanted to maintain logical coherence and ensure consistency between different sections of the manuscript, trying to make it easier for readers to understand our work. Here, to demonstrate the repeatability of CPD, we have additionally provided R-T data for 5 samples of different sizes (Fig. R2). Among them, sample A has dimensions of $8 \times 4 \text{ mm}^2$, sample B has dimensions of $4 \times 4 \text{ mm}^2$, sample C has dimensions of $8 \times 5 \text{ mm}^2$, sample D has dimensions of $10 \times 5 \text{ mm}^2$, and sample E has dimensions of $5 \times 5 \text{ mm}^2$. It can be observed that there are slight differences in resistance values among samples of different sizes. On one hand, the differences in resistance are expected due to the varying sizes of the samples. On the other hand, manual operations inherently introduce various unavoidable sources of error, so these variations are understandable. Furthermore, such differences are common in NTC materials. Taking the renowned Cernox temperature sensor as an example, the resistance values of different samples at the same temperature can vary by up to 2 orders of magnitude. Furthermore, their other

characteristics, including the negative temperature coefficient, low-temperature behavior, cyclic stability, and fitting characteristics, all align with the conclusions presented in the manuscript. Therefore, we believe that this is sufficient to demonstrate the repeatability of CPD samples.

Fig. R2. R-T curves and of CPD samples with different sizes: sample A ($8 \times 4 \text{ mm}^2$), sample B ($4 \times 4 \text{ mm}^2$), sample C ($8 \times 5 \text{ mm}^2$), sample D ($10 \times 5 \text{ mm}^2$), and sample E ($5 \times 5 \text{ mm}^2$).

Lastly, from a conceptual standpoint, I find myself in disagreement with the authors' motivation for temperature measurement. The manuscript refers to "traditional thermometers being limited to temperatures above 20 mK" based on Ref. 32. However, this limitation does not stem from the thermometer materials employed in Ref. 32. Instead, it arises from the technical challenges associated with the dilution refrigerator used in Ref. 32. Although dilution refrigerators can achieve temperatures in the millikelvin range, reaching this level is not easy. Moreover, the abstract states, "Currently, millikelvin temperatures and below are measured through the characterization of a certain thermal state of the system as there is no traditional thermometer capable of measuring temperatures at such low levels." Yet, the approach of characterizing a certain thermal state of the system (presumably indicating a quantum gas) for temperature measurement is entirely reasonable. This technique circumvents the heat capacity issue, wherein the thermometer's heat capacity could influence temperature measurements for samples with small heat capacities or at extremely low temperatures.

Response: Thank you for the reviewer's constructive suggestions. Cryogenic

technology and cryogenic temperature measurement technology are two mutually developing technologies. As pointed out by the reviewer, most current dilution refrigerators face challenges in reaching temperatures below 20mK. However, this does not imply that the development of sensors capable of measurements in even lower temperature ranges is meaningless. On the contrary, to achieve lower temperatures and maintain stable over extended periods, excellent temperature measurement components are indispensable. Taking the example of the known lowest temperature dilution refrigerator, it can reach temperatures below 2 mK^{1,2}. Nevertheless, calibrating or testing this equipment within the temperature range of 10 mK and below requires the use of five 47 Ω Speer resistors and three vibrating wire resonators, which belong to two separate temperature measurement systems. Such complex temperature measurement system not only increases the difficulty of calibration but also elevates the overall equipment cost. Furthermore, as a highly consumable component, cryogenic temperature sensor usually elevates the long-term operational costs of the device. Therefore, simple, cost-effective, and operable at even lower temperatures cryogenic temperature sensors undoubtedly hold great promise for the advancement and practical application of cryogenic temperature technology. By a simple cost calculation between the CPD and the RuO₂ mentioned in reference 13, it can be observed that artificially synthesized diamond powder, at 25 carats (5g), typically costs between 10-20 US dollars, whereas the same weight of precious metal oxide, RuO₂, can be priced at 300 dollars or even more. Furthermore, the process only requires a high-temperature sintering under Ar atmosphere, which can be achieved by most of the high-temperature tube furnaces available in the market. Therefore, even without considering measurements at lower temperatures, CPD proves to be highly competitive as an alternative to existing cryogenic temperature sensors available in the market.

In addition, as the reviewer has mentioned, low-temperature sensing requires minimizing the impact of sensor heat capacity on the environmental temperature. CPD, like diamond, exhibits low heat capacity and low thermal expansion coefficient, as shown in Fig. R3. According to the Debye model², as the temperature approaches absolute zero, the heat capacity of CPD also approaches zero. Based on experimental values, at 25 K, CPD's heat capacity is already less than 0.01 J·K⁻¹·mol⁻¹. This means that at lower temperatures, CPD becomes more sensitive to temperature changes. At the same time, the resistance plays a crucial role in determining the self-heating effect of the temperature sensor. The higher the resistance, the more severe the self-heating effect

of the temperature sensor. Therefore, CPD's low heat capacity and low resistance can effectively reduce the self-heating effect during temperature measurements.

Fig. R3. Thermal properties of CPD. a, The thermal expansion coefficient of CPD. b, The heat capacity of CPD.

In conclusion, even though sub-20 mK temperature control technologies are not yet widespread, we do think CPD holds the potential to drive the development of cryogenic temperature technologies and reduce their associated costs.

On this note, I sense that the present paper might not have thoroughly considered the practical implementation of this "thermometer material" as an actual thermometer. Given my reservations concerning the quality of the data and analysis, I hold significant concerns regarding the suitability of publishing this work in Nature Communications, a journal that necessitates broad interest.

References

1. Zu, H., Dai, W. & Waele, A. T. A. M. De. Development of dilution refrigerators - A review. *Cryogenics*, **121**, 103390 (2022).
2. Cousins, D. J. et al. An Advanced Dilution Refrigerator Designed for the New Lancaster Microkelvin Facility. *J. Low Temp. Phys.* **114**, 547–570 (1999).
3. Vasil'ev, O. O., Muratov, V. B. & Duda, T. I. The study of low-temperature heat capacity of diamond: Calculation and experiment. *J. Superhard Mater.* **32**, 375-382 (2010).

REVIEWERS' COMMENTS

Reviewer #1 (Remarks to the Author):

After carefully read the revised manuscript many times, the authors have satisfactorily addressed my concerns. In addition, in my opinion, they have also provided solid evidence to address most concerns from reviewers 2# and 3#. I suggest this interesting work is worth for publication in Nature Communications.

Reviewer #2 (Remarks to the Author):

The paper has been improved a lot and the authors had addressed all issues I raised in the first review. I have no further comments to the revised manuscript and recommend publication in Nature Communications except that the word of "DPD" should be replaced by "CPD" in Fig. 4.

Reviewer #3 (Remarks to the Author):

I think The authors successssfully have addressed concerns raised by the reviewers. The additional experimental verification using dilution refrigerator supports the paper's claim well, by which I can recommend its publication. I have some minor comments as follows.

1. The order of supplementary figure can be changed. It looks that Suppl. Fig. 7 appears earlier than Suppl. Fig. 6. Please check it again.
2. It looks that Suppl. Fig. 9 is not mentioned in the main text. Am I correct?
3. Fitting results tables inserted in Fig. 2a, c, use very small letters. If all the parameters should be shown, type them in the captions or prepare additional tables in Supplementary Information.

RESPONSE TO REVIEWERS' COMMENTS

The authors have made point-by-point responses (marked as **blue** colour) to each comment (marked as **black** colour).

Reviewer #1's comments:

After carefully read the revised manuscript many times, the authors have satisfactorily addressed my concerns. In addition, in my opinion, they have also provided solid evidence to address most concerns from reviewers 2# and 3#. I suggest this interesting work is worth for publication in Nature Communications.

Answer: Thank you for the reviewer. We would like to express our gratitude once again for the valuable suggestions provided by the reviewer, which have greatly benefited us.

Reviewer #2's comments:

The paper has been improved a lot and the authors had addressed all issues I raised in the first review. I have no further comments to the revised manuscript and recommend publication in Nature Communications except that the word of “DPD” should be replaced by “CPD” in Fig. 4.

Answer: Thank you for providing the modification suggestions. The above issue has been revised in the manuscript. We would like to express our gratitude once again for the valuable suggestions provided by the reviewer, which have greatly benefited us.

Reviewer #3's comments:

I think The authors successssfully have addressed concerns raised by the reviewers. The additional experimental verification using dilution refrigerator supports the paper's

claim well, by which I can recommend its publication. I have some minor comments as follows.

1. The order of supplementary figure can be changed. It looks that Suppl. Fig. 7 appears earlier than Suppl. Fig. 6. Please check it again.

2. It looks that Suppl. Fig. 9 is not mentioned in the main text. Am I correct?

3. Fitting results tables inserted in Fig. 2a, c, use very small letters. If all the parameters should be shown, type them in the captions or prepare additional tables in Supplementary Information.

Answer: Thank you for providing the modification suggestions. The above issues have been revised in the manuscript and supplementary information. We would like to express our gratitude once again for the valuable suggestions provided by the reviewer, which have greatly benefited us.